# Geometric and Physical Constraints Synergistically Improve Neural PDE Surrogates

## Abstract

Neural PDE surrogates can improve on cost-accuracy tradeoffs of classical solvers, but often generalize poorly to new initial conditions, accumulate errors over time. To close the performance gap between training and long-term inference, we constrain neural surrogates with symmetry equivariance and physical conservation laws as hard constraints, using novel input and output layers that support scalar and vector fields on the staggered grids commonly used in computational fluid dynamics. We systematically investigate how these constraints affect accuracy, individually and in combination, on two challenging tasks: shallow water equations with closed boundaries and decaying incompressible turbulence. Compared to a strong baseline, both types of constraints improve performance consistently across autoregressive prediction steps, accuracy measures, and network sizes. Symmetries are more effective but do not make physical constraints redundant. Doubly-constrained surrogates were more accurate for the same network and dataset sizes, and generalized better to initial conditions and durations beyond the range of training data.

## 1 Introduction

Recently, neural networks have shown promising results in predicting the time evolution of PDE systems, often achieving cost-accuracy tradeoffs that outperform traditional numerical methods (Li et al., 2020; Gupta & Brandstetter, 2022; Stachenfeld et al., 2021; Takamoto et al., 2022; Long et al., 2019; Um et al., 2020; Kochkov et al., 2021). However, obtaining accurate and stable autoregressive 'rollouts' over long durations remains notoriously difficult. Several techniques have been proposed to address this, including physical constraints, symmetry equivariance, time-unrolled training, specialized architectures, data augmentation, addition of input noise and generative modelling (Sanchez-Gonzalez et al., 2020; Lippe et al., 2024; Stachenfeld et al., 2021; Kohl et al., 2024; Brandstetter et al., 2022a; Fanaskov et al., 2023; Bergamin et al., 2024; Sun et al., 2023; Hsieh et al., 2019; Tran et al., 2021; Li et al., 2023; Bonev et al., 2023). Nonetheless, the relative effectiveness of these strategies remains largely ambiguous, and transparent, systematic comparisons remain elusive.

Here we systematically investigate the utility of symmetry constraints and physical conservation laws, alone and in combination. Across multiple tasks, accuracy measures and scenarios, we show a clear, reproducible and robust benefit from these constraints, and find they can be combined synergistically. In order to apply them broadly, we introduce novel input and output layers extending these inductive biases for the first time to staggered grids.

## 2 Background and Related Work

**Neural PDE surrogates**   We aim to train neural networks to predict the time evolution of a system of PDEs. We consider time-dependent variable fields $\boldsymbol{w}(t,x) \in \mathbb{R}^m$, for $x \in \Omega \subset \mathbb{R}^d$, $t \in [0,T]$ and

$$\frac{\partial \boldsymbol{w}}{\partial t} = \mathcal{F}(t, \boldsymbol{x}, \boldsymbol{w}, \nabla \boldsymbol{w}, \nabla^2 \boldsymbol{w}, \ldots) \tag{1}$$

Starting from initial conditions (ICs) $\boldsymbol{w}(0, \boldsymbol{x})$ and boundary conditions (BCs) $B[\boldsymbol{w}](t, \boldsymbol{x}) = 0, \forall x \in \partial\Omega$, the solution can be advanced with a fixed time step:

$$\boldsymbol{w}(t + \Delta t, \cdot) = \mathcal{G}[\boldsymbol{w}(t, \cdot)], \tag{2}$$

where $\mathcal{G}$ is an update operator. To provide training data and evaluate performance we use a reference solution generated by a numerical solver with space- and time-discretized variable fields.

Recent studies have trained neural surrogates to approximate $\mathcal{G}$ (Greenfeld et al., 2019; Gupta & Brandstetter, 2022; List et al., 2024; Lippe et al., 2024; Li et al., 2020; Tripura & Chakraborty, 2023; Raonic et al., 2024). The neural network can also be combined with a numerical solver, in so-called 'hybrid methods' (Bar-Sinai et al., 2019; Tompson et al., 2017; Kochkov et al., 2021; Bukka et al., 2021; Long et al., 2019).

A major challenge remains training neural surrogates to give stable and accurate results over long autoregressive rollouts. Several techniques have been proposed, including physical constraints, symmetry constraints, training with input noise, unrolled training and generative modelling. However, a clear consensus on the relative effectiveness of these approaches remains elusive, and applying them in various tasks is not always straightforward.

**Symmetry equivariance**  Suppose $f : \boldsymbol{w} \to \boldsymbol{z}$ is an operator mapping between two multidimensional variable fields $\boldsymbol{w}(\boldsymbol{x}), \boldsymbol{z}(\boldsymbol{x})$ defined on $\Omega \subset \mathbb{R}^d$. Then for a group $G$ of invertible transformations on $\mathbb{R}^2$, $f$ is *equivariant* if it commutes with the actions of $G$ on $\boldsymbol{w}$ and $\boldsymbol{z}$. That is, there should exist transformations $\mathcal{T}_g, \mathcal{T}_g'$ operating on $\boldsymbol{w}, \boldsymbol{z}$ respectively, such that

$$[f \circ \mathcal{T}_g \boldsymbol{w}](\boldsymbol{x}) = [\mathcal{T}_g' \circ f \boldsymbol{w}](\boldsymbol{x}), \qquad \forall g \in G, \boldsymbol{x} \in \Omega \tag{3}$$

When $w$ is a scalar field, $\mathcal{T}, \mathcal{T}'$ simply resamples it at coordinates defined by the action of $G$ on $\mathbb{R}^d$

$$\mathcal{T}_g^{\text{scalar}} w(\boldsymbol{x}) = w(g^{-1}\boldsymbol{x}) \tag{4}$$

Other field types transform in more complex ways. For example, the action of a $90°$ rotation $R$ on a 2D vector field both resamples the field and rotates each vector:

$$\mathcal{T}_R^{\text{vector}}[w_1(\boldsymbol{x}), w_2(\boldsymbol{x})] = [-w_2(R^{-1}\boldsymbol{x}), w_1(R^{-1}\boldsymbol{x})] \tag{5}$$

The range of possible actions is described by $G$'s group representations. Efficient, full-featured software packages exist for equivariant convolutions (Cesa et al., 2022) and self-attention (Romero & Cordonnier, 2020), and have proven useful in image classification (Chidester et al., 2019) and segmentation (Veeling et al., 2018). Equivariance has been used to improve neural PDE surrogates in some cases (Wang et al., 2020; Helwig et al., 2023; Smets et al., 2023; Huang & Greenberg, 2023; Ruhe et al., 2024). Numerical integration methods can also benefit from maintaining PDE symmetries Rebelo & Valiquette (2011).

We restrict ourselves to discrete symmetry groups on regular grids, though some approaches for continuous symmetries have been proposed (Weiler & Cesa, 2019; Cesa et al., 2022). We note that standard convolutions and self-attention with relative encoding are already equivariant to translations (up to boundary effects).

**Staggered grids**  Fluid dynamical systems are often simulated using staggered grids (Fig. 1, left), in which variables such as pressure, density, divergence or velocity along each axis are represented at different locations. This approach can avoid grid-scale numerical artifacts in numerical integration, and is common in fluid dynamics (Holl & Thuerey, 2024; Kochkov et al., 2021; Jasak, 2009; Stone et al., 2020) as well as atmospheric (Jungclaus et al., 2022; De Pondeca et al., 2011) and ocean models (Korn et al., 2022; Madec et al., 2023). Unfortunately, existing equivariant network layers (Cesa et al., 2022; Romero & Cordonnier, 2020) assume $\mathcal{T}_g$ can be described by a resampling operation followed by an independent transformation at each grid point as in Equation 5, but on staggered grids rotation and reflection do not take this form.

**Physical constraints**  Neural surrogates have frequently been applied to physical systems, many of which include known conservation laws. To improve accuracy, stability, and generalization capabilities, these laws can be imposed through additional loss terms (Read et al., 2019; Wang et al., 2020; Stachenfeld et al., 2021; Sorourifar et al., 2023). Taking the strategy of physics-derived loss terms to its ultimate limit, on arrives at unsupervised training on PDE-derived losses for discretized (Wandel et al., 2020; Michelis & Katzschmann, 2022) or continuous solutions (Raissi et al., 2019). Alternatively, one can reparameterize network outputs to respect hard constraints (Mohan et al., 2020; Beucler et al., 2021; Chalapathi et al., 2024; Cranmer et al., 2020; Greydanus et al., 2019). Here we focus on discretized, supervised approaches, which have proven more competitive in larger and more complex PDE systems (Takamoto et al., 2022).

## 3 SYMMETRY- AND PHYSICS-CONSTRAINED NEURAL SURROGATES

In this work, we assess the separate and combined benefits of symmetries and conservation laws for neural PDE surrogates. To achieve this, we construct specialized input layers that support equivariance on staggered grids (Fig. 7), as well as output layers that enforce both equivariance and conservation laws. When comparing to non-equivariant networks, we replace equivariant convolutions using standard convolutions with the same size and padding options, adjusting channel width to match total parameter counts (details in Appendix B).

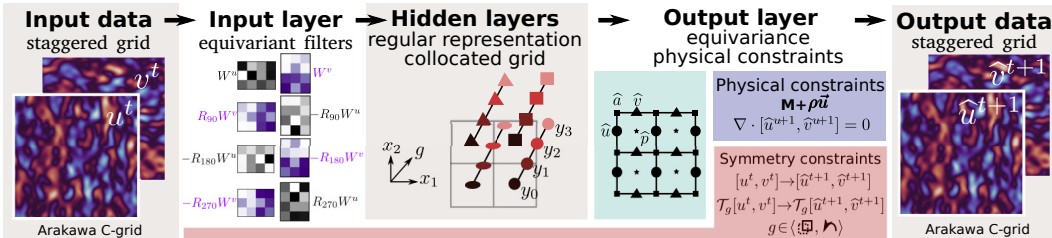

Figure 1: Symmetry- and physics-constrained neural surrogate for incompressible flow on a staggered grid. A rotation-equivariant input layer maps velocities onto an unstaggered regular representation, hidden layers employ steerable convolutions and the equivariant output layer enforces conservation laws on mass and momentum ($m + \rho\vec{u}$) as it maps to staggered velocities.

Fig. 1 demonstrates our overall framework for constructing equivariant, conservative neural surrogates. As an illustrative example, we show the incompressible Navier Stokes equations, with equivariance to translation and rotation, momentum conservation and a divergence-free condition (equivalent to mass conservation). Input data defined on staggered grids are mapped through novel equivariant input layers to a set of convolutional output channels defined at grid cell centers. Each channel of internal activations is *regular representations*: a group of channels indexed by $G$,[1] on which $G$ acts by transforming each spatial field and by permuting the channels according to the group action Cohen & Welling (2016); Cesa et al. (2022). Essentially, regular representations are real-valued functions of the discrete symmetry group $G$. This formulation allows us to use the preexisting library escnn (Cesa et al., 2022) for all internal linear transformations between hidden layers. Finally, we employ novel output layers to map the regular representation back to the staggered grid, while simultaneously enforcing conservation laws as hard constraints.

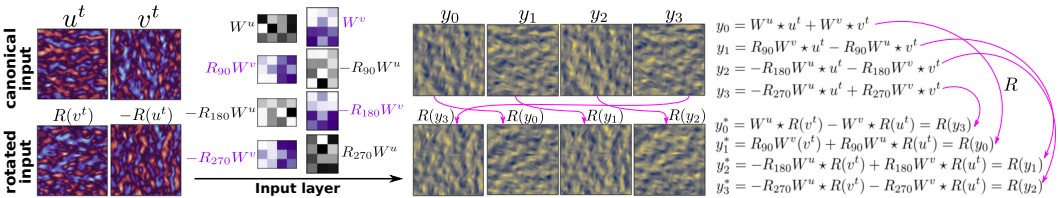

Figure 2: Action of rotation-equivariant input layer on staggered velocity fields (top left). The filter bank is transformed by each $g \in G$ to compute a $G$-indexed regular representation $y$. Rotation-transforming inputs (bottom left) yields permuted, rotated output channels.

**Input layers** We consider input data on staggered Arakawa C-grids (Fig. 1 left, Fig. 7). This grid consists of square cells, where variable fields can be defined at cell centers (typically scalar fields like pressure, surface height, or divergence), at the midpoints of cell interfaces (such as velocity components) at vertices (e.g. stream functions). For an $n \times n$ 2D grid of cells, there is an $(n+1) \times n$ grid of interfaces in the $x_1$ direction (along rows, including boundaries), and an $n \times (n+1)$ grid of interfaces in the $x_2$ direction (along columns).

We designed convolutional input layers to take scalar inputs at cell centers and/or vector fields with components defined at interfaces. Inputs at interfaces are first processed with a bank of convolutional

---

[1]Technically, channels of the regular representation are indexed by the non-translational subgroup of $G$.

filters, which are each even-sized along the coordinate axis orthogonal to a single set of interfaces and odd-sized along all other axes (Fig. 2, left). This filter bank is *collectively* transformed according to each element of the symmetry group $G$, while being applied to the input data. Note that, similar to the transformation of vector fields (Eq. 5), these filter banks undergo collective transformation by rotations and reflections, not only through resampling, but also through permutation and sign flips (Fig. 2, right). When we rotation-transform input vector fields (Eq. 5), this has the effect of permuting and rotating the outputs of our input layer, as required for an equivariant mapping onto a regular representation (fig. 2, magenta arrows). Inputs at cell centers are processed with separate, standard equivariant convolution layers. Convolutions for both interface and center-defined input variables produce regular representation outputs, which are then combined to compute the total input to the network's first hidden layer. We provide implementations of 2D input layers for translation-rotation (p4) and translation-rotation-reflection (p4m). Further details on input layers can be found in Appendix C.

**Output layers**  We designed convolutional output layers mapping from regular representations to staggered C-grid variables (Fig. 1, center-right). As for the input layers, we use separate convolutional filter banks for cell- and interface-centered variables, but now additionally support vertex-centered outputs scalar for the purpose of enforcing physical constraints (see below). Scalar face-centered outputs are computed using pooling layers over a regular representation (Cohen & Welling, 2016). Vector field outputs at each cell interface are computed as linear combinations of regular representations at the surrounding two cell centers, with constraints imposed on the weights to satisfy the equivariant transformation of vector fields (Eq. 5, details in D). Vertex-centered scalar outputs are computed using even-size square filters, followed by pooling layers operating over $G$-indexed channels.

**Conservation laws**  Conservation laws for scalar quantities defined at cell centers and vectors at cell interfaces are imposed through global mean corrections (details and alternatives in appendix E). As conservation of mass for incompressible flow is equivalent to a divergence free condition, we enforce this by training the network to output a scalar stream function $a$ at cell vertices, and follow Wandel et al. (2020) in defining

$$[\widehat{u}^{t+1}, \widehat{v}^{t+1}] = \nabla \times [0, 0, a] \tag{6}$$

As the Helmholtz-Hodge decomposition of a vector field consists of curl-free and divergence-free components, eq. 6 guarantees the learned vector field is divergence free, and that any divergence free vector field can be represented in this way. For an $n \times n$ grid, periodically padding $a$ to $(n+1) \times (n+1)$ guarantees momentum conservation.

## 3.1 BASE ARCHITECTURE

In order to measure the efficacy of symmetry constraints and conservation at the cutting edge of neural PDE surrogate research, it was essential to choose a flexible base architecture with efficient training and inference that has produced highly competitive results. To this end we selected the "modern U-net" introduced in Gupta & Brandstetter (2022), which modifies the original U-net (Ronneberger et al., 2015) for improved performance as a PDE surrogate. This architecture has shown strong results in Kohl et al. (2024), and a similar version performed well in Lippe et al. (2024). We used this architecture without self-attention layers, which did not significantly affect our results.

## 3.2 TRAINING

We trained neural surrogates using an MSE loss $\mathcal{L} = \frac{1}{N} \left\| \widehat{\boldsymbol{w}}^{t+1} - \boldsymbol{w}^{t+1} \right\|_2^2$, where $N$ is the number of discretized PDE field values. All data fields were normalized by subtracting the mean and dividing by the standard deviation, with common values for both components of vector fields. We trained on 8 A100 GPUs with the ADAM optimizer (Kingma, 2014), batch size 32 and initial learning rate 1e-4. We employed early stopping when validation loss did not reduce for 10 epochs, and accepted network weights with the best validation loss throughout the training process.

# 4 PDE SYSTEMS

We considered two challenging 2D fluid dynamical PDEs, with the same staggered grid and symmetries but different variables, BCs/ICs, reference solvers and conservation laws. Full sets of constraints for each system and names for each combination appear in tables (1-2), while PDE parameters and further numerical details appear in tables (3-4).

Table 1: Geometric and physical constraints for SWEs

| Conservation laws | Symmetries | | |
|---|---|---|---|
| | ⊡ | ⊡∩ | ⊡∩⬙ |
| None ∅ | **p1**/∅ | **p4**/∅ | **p4m**/∅ |
| Mass **M** | **p1**/M | **p4**/M | **p4m**/M |

Table 2: Geometric and physical constraints for INS

| Conservation laws | Symmetries | | |
|---|---|---|---|
| | ⊡ | ⊡∩ | ⊡∩⬙ |
| None ∅ | **p1**/∅ | **p4**/∅ | **p4m**/∅ |
| Momentum $\rho\vec{u}$ | **p1**/$\rho\vec{u}$ | **p4**/$\rho\vec{u}$ | **p4m**/$\rho\vec{u}$ |
| Mass/moment. **M**+$\rho\vec{u}$ | **p1**/M+$\rho\vec{u}$ | **p4**/M+$\rho\vec{u}$ | **p4m**/M+$\rho\vec{u}$ |

## 4.1 CLOSED SHALLOW WATER SYSTEM

The shallow water equations (SWEs) are widely used to describe a quasi-static motion in a homogeneous incompressible fluid with a free surface. We consider nonlinear SWEs in momentum- and mass conservative form on domain $\Omega$ with 'closed' Dirichlet BCs (Song et al., 2018):

$$\frac{\partial \boldsymbol{u}}{\partial t} = -C_D \frac{1}{h}\boldsymbol{u}|\boldsymbol{u}| - g\nabla\zeta + a_h\nabla^2\boldsymbol{u}; \qquad \frac{\partial \zeta}{\partial t} = -\nabla \cdot (h\boldsymbol{u}) \qquad\qquad \text{on } \Omega \qquad (7)$$

$$\boldsymbol{u} = \boldsymbol{0} \qquad\qquad\qquad\qquad \text{on } \partial\Omega \qquad (8)$$

where $\zeta$ is fluid surface elevation, $\boldsymbol{u} = [u, v]$ is the velocity field, $d$ and $h$ respectively represent the undisturbed- and disturbed fluid depth (so that $h = d + \zeta$) and $\partial\Omega$ is a closed domain boundary. $a_h$ is the horizontal turbulent momentum exchange coefficient, $C_D$ is the bottom drag coefficient and $g$ is gravitational acceleration. SWE simulations exhibit travelling waves that reflect from domain boundaries, temporarily increasing in height as they self-collide. This system is fundamentally more challenging than previously proposed SWE-based benchmarks with open (Takamoto et al., 2022) or periodic BCs (Gupta & Brandstetter, 2022), due to the combination of self-interfering wave patterns, incompressibility and altered dynamics at pixels near the domain boundaries.

**Numerical reference solution** Closed BCs and incompressibility lead to stiff dynamics, so explicit solvers are inefficient. Instead, we generate data using a semi-implicit scheme (Backhaus, 1983) that represents $\zeta$ and $[u, v]$ on a staggered Arakawa C-grid (Arakawa, 1977) and solves a sparse linear system at each time step $\Delta t = 300s$.

Grids are $100 \times 100$, $100 \times 99$, and $99 \times 100$ respectively for $\zeta$, $u$, and $v$. We trained on 50 simulations spanning 50 h (600 time steps) each. ICs were $\zeta = 0$ except for a 0.1 m high square-shaped elevation, and $[u, v] = 0$. The square had side length uniformly distributed from 2-28 grid cells and random position. Simulations in Fortran required 67 seconds on the CPU. Testing and validation data included 10 simulations. Surrogates used the same time step as the solver.

**Symmetries and conservation laws** The shallow water system in Eqs. 7-8 is equivariant to rotations and reflections. Since boundary effects interfere with translation equivariance, we provide a boundary mask as an additional input channel. These symmetries are illustrated and empirically verified in Fig. 7. The only conserved quantity for SWE is mass (defined as $\Delta x^2 h$ times fluid density, so that the mean of $\zeta$ is also conserved). Momentum is not conserved due to reflection of waves from the closed boundaries.

## 4.2 DECAYING TURBULENCE

The incompressible Navier–Stokes equations (INS) describe momentum balance for incompressible Newtonian fluids. Our 2D version relates velocities $\boldsymbol{u} = [u, v]$ to pressure $p$:

$$\frac{\partial \boldsymbol{u}}{\partial t} + (\boldsymbol{u} \cdot \nabla)\boldsymbol{u} = -\frac{\nabla p}{\rho} + \mu\nabla^2\boldsymbol{u}; \qquad \nabla \cdot \boldsymbol{u} = 0 \qquad (9)$$

where $\rho$ fluid density and $\mu$ is kinematic viscosity. Here we consider the 'decaying turbulence' scenario introduced by Kochkov et al. (2021). The velocity field is initialized as filtered Gaussian noise containing high spatial frequencies. Predicting the evolution of the velocity field is challenging, since eddy size and Reynolds number change over time as structures in the flow field coalesce, and the velocity field becomes smoother and mroe uniform over time.

**Numerical reference solution** We solve eq. 9 with C-grid staggering of velocities, using `jax-cfd` (Kochkov et al., 2021; Dresdner et al., 2022). We follow the data generation setup of Kochkov et al. (2021) and Stachenfeld et al. (2021). The solver used a grid of $576 \times 576$ cells and a $44$ ms time step over $224$ seconds. Training data were coarsened to a time step of $0.84$ s, and resolution was reduced to $48 \times 48$ (Stachenfeld et al., 2021) using face-averaging to conserve momentum and the divergence-free condition. The solver used a standard pressure projection approach, so that $p^{t+1}$ is computed along with $\vec{u}^{t+1}$ along with $\vec{u}^{t}$, and $p^t$ is discarded. We use a burn-in of 148 coarsened steps, leaving 120 steps for training. We trained on 100 ICs consisting of filtered Gaussian noise with peak spectral density at wavenumber 10 (that is, 10 cycles across the spatial domain). We used 10 initial conditions for testing and validation.

**Symmetries and conservation laws** INS in Equation 9 are equivariant to translations, rotations and reflections, as illustrated and empirically verified in Figure 9. Conserved quantities include momentum (equivalent to a constant mean for each velocity component, since $\rho$ is constant), and mass (manifested here as the divergence-free condition on the velocity field).

## 5 RESULTS

### 5.1 CLOSED SHALLOW WATER SYSTEM

We first trained and evaluated neural surrogates for the SWE system. For this task, we followed a hybrid learning strategy, based on the observation that the semiimplicit numerical integration scheme calculates $\zeta^{t+1}$ slowly with an iterative solver, but then calculates $[u^{t+1}, v^{t+1}]$ given $\zeta^{t+1}$ quickly and trivially through a mathematical formula. We therefore trained surrogates to predict only $\widehat{\zeta}^{t+1}$, and calculated $[\widehat{u}^{t+1}, \widehat{v}^{t+1}]$ as in the numerical solver (Appendix G). Keeping parameter counts constant, we compared networks trained equivariant to 3 symmetry groups: p1 (translation only), p4 (translation-rotation) and p4m (translation-rotation-reflection). We also compared mass conserving networks (m) to those without physical constraints ($\varnothing$). Table 1 lists all constraint combinations used for training, which took 0.5 h for non-equivariant networks and 2h for equivariant networks on an A100 GPU.

Figure 3a compares autoregressive rollouts from unconstrained (p1/$\varnothing$) and maximally constrained networks (p4m/M). p4m/M maintained accurate results for a much greater time interval, and in this case was visually indistinguishable from the reference solution throughout the simulation (results for all networks are shown in 13). Over 20 random held-out ICs in this test, p4m/M exhibited lower normalized RMSE values and high correlations for predicted $\zeta$ values than other networks (Figure 3b-c)). We also compared to unconstrained networks trained with input noise (p1m/$\varnothing + \varepsilon$, details in 14), which improved long-rollout performance as previously proposed (Stachenfeld et al., 2021; Lippe et al., 2024), but was not as effective as the combination of symmetry constraints and conservation laws. Compared to other networks, p4m/M was able to train for more epochs before early stopping occurred, and achieved a validation lower loss (Fig. 3d). It also achieved lower values over a greater fraction of held-out ICs (Fig. 3e), maintained energy conservation (which was not constrained by any archicture) for longer (Fig. 3f) and stayed correlated to the reference solution for longer (Fig. 3g). Overall, we found that symmetry constraints were more effective than conservation laws, but that the benefits provided by each combined synergistically, rather than redundantly.

### 5.2 DECAYING TURBULENCE

We next trained and evaluated neural surrogates for INS. Here we used the velocity fields $[u, v]$ as both inputs and outputs. As for SWEs, with consider p1, p4 and p4m equivariance, but now considered 3 levels of physical constraints: unconstrained ($\varnothing$) conservation of momentum ($\rho\vec{u}$) and conservation of both momentum and mass ($m + \rho\vec{u}$). Table (2) lists all constraint combinations used

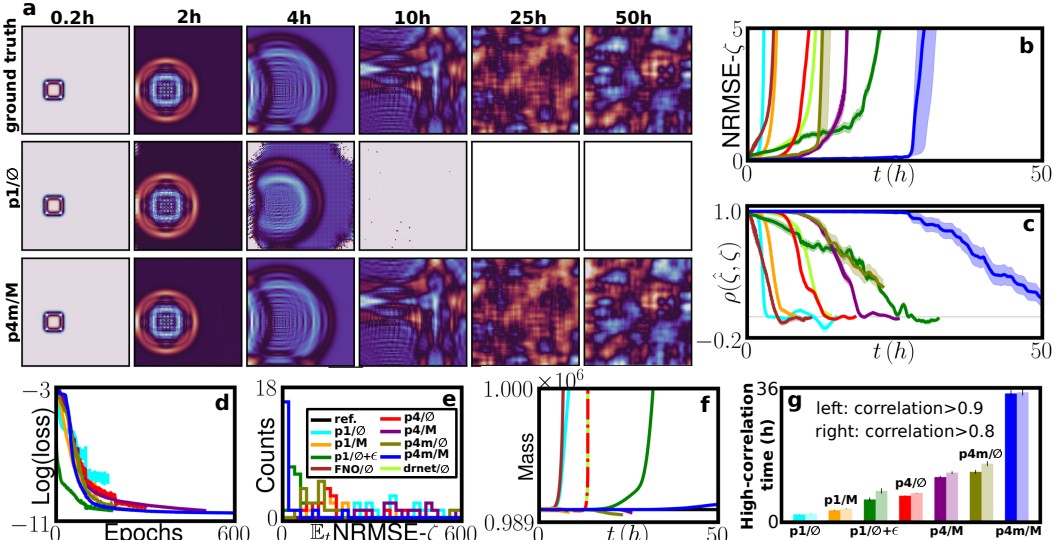

Figure 3: p4m/M (symmetry+physics constraints) outperforms other networks with similar parameter counts on SWE. (a) Reference surface disturbance $\zeta$ with predictions from p1/$\varnothing$ and p4m/M. (b-c) Accuracy over 50h rollouts, with standard error of the mean over 20 ICs. (d) Training loss over iterations. (e) Histogram of $\mathbb{E}_t$NRMSE over 20 ICs. (f) Violation of mass conservation for all methods (black line shows reference simulation). (g) High correlation times for each model.

for training, which took 0.4 h for nonequivariant networks and 1.4 h for equivariant networks on an A100 GPU.

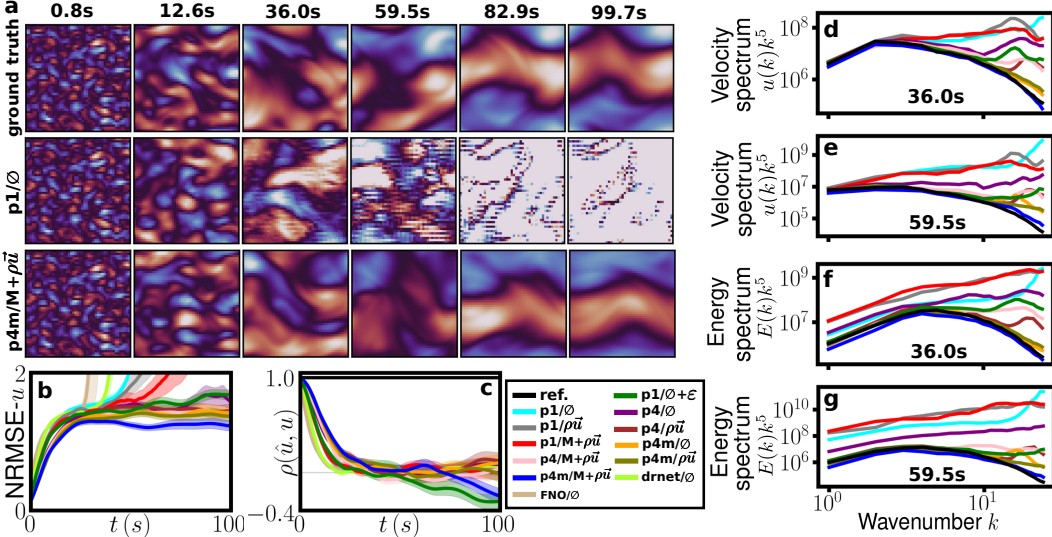

Figure 4: p4m/M+$\rho\vec{u}$ outperforms other networks with similar parameter counts on INS. (a) Reference horizontal velocity with predictions from p1/$\varnothing$ and p4m/M+$\rho\vec{u}$. (b-c) Accuracy over 50h rollouts, with standard error of the mean over 30 ICs. (d-e) Log-log plots of the average velocity power spectrum from 30 ICs ar $t = 36.0s, 59.5s$. Spectra measure the strength of the chaotic field's features for each wavenumber $k$ (number of cycles across the domain). (f-g) Comparison of the energy spectrum from all methods to the ground truth. Both the velocity and energy spectra p4m/M+$\rho\vec{u}$ align best with the reference. Spectra are scaled by $k^5$.

Figure (4-a) compares autoregressive rollouts from unconstrained (p1/$\varnothing$) and maximally constrained networks (p4m/M+$\rho\vec{u}$). As for the SWEs, we observed improvements to accuracy and stability of

INS surrogates for both types of constraints (fig. 4b-c), and best results for maximally constrained networks, which also outperformed networks trained with the same input noise used for this task in Stachenfeld et al. (2021). Unconstrained networks were particularly susceptible to numerical instability in this task (for all networks' rollouts, see fig. 16-17).

To evaluate the performance of neural surrogates beyond the time at which their predictions decorrelate from the reference solution, we followed previous studies (Kochkov et al., 2021; Lippe et al., 2024; Stachenfeld et al., 2021) in further comparing the power spectra of predicted velocity fields, and of energy fields $\frac{1}{2}|\vec{u}|^2$, to those of the reference solver. Even after average correlation with the reference solution reached 0, we found that p4m/M+$\rho\vec{u}$ networks matched the spectra of the reference solver far better than all other methods, consistently across multiple rollout times and especially at the highest spatial frequencies (fig. 4d-g, additional spectra in fig. 18). We also trained p4m/M+$\rho\vec{u}$ networks with input noise, resulting in lower accuracy but excellent long-term numerical stability (fig. 19).

### 5.3 GENERALIZATION

We next evaluated how physical and symmetry constraints affect generalization performance of neural surrogates to ICs outside their training sets.

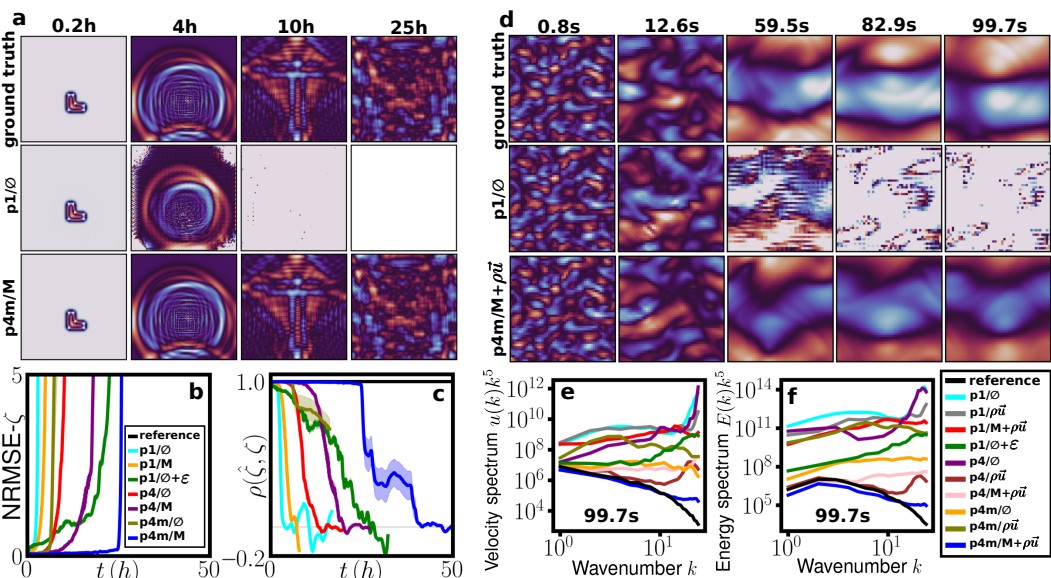

Figure 5: Generalization beyond training data. (a) SWE rollouts from p1/∅ p4m/M on L-shaped ICs. (b-c) Accuracy of each network over six generalization tests (appdendix H.3). (d) INS rollouts from p1/∅ and p4m/M+$\rho\vec{u}$ on ICs with peak wavenumber of 8. (e-f) Velocity- and energy spectra for INS at $t = 99.7$ s, averaged over 10 ICs.

**Closed Shallow Water System** We tested on ICs defined as a sum of two rectangular elevations 0.1 m in height, with randomly varying location and shape (details in 13). Fig. 5-a shows an example in which these rectangles have combined to form an 'L' shape. As previously, we found the maximally constrained model p4m/M to outperform alternatives with equal parameter counts (Fig. 5b-c). Additional generalization results are show in Figs. 21-22.

**Decaying Turbulence** We tested surrogates on ICs with peak wavenumber changed from 10 to 8 or 6. p4m/M+$\rho\vec{u}$ rollouts more closely matched the reference solver (Fig. 5d) and its spectra (Fig. 5e-f). Additional generalization results are shown in Fig. 24.

## 5.4 EFFECTS OF NETWORK AND DATASET SIZE

We further investigated how the enhancement of neural INS surrogates by symmetry physical constraints depends on the size of the network and dataset. We trained p1/∅ and p4m/M+$\rho\vec{u}$ networks with 0.1M, 2M and and 8.5M parameters on the same dataset (100 simulations). Evaluating predictions at 4.2 and 12.6 s, we observed lower errors and high correlations for p4m/M+$\rho\vec{u}$ at both times and all three network sizes. The relative improvement of brought about by p4m/M+$\rho\vec{u}$ over p1/∅ was greatest for smaller networks and for longer forecast horizons, and overall performance was best for larger networks.

Training 0.1M-parameter p1/∅ and p4m/M+$\rho\vec{u}$ networks on datasets of 100, 400, and 760 simulations showed that constraints enhanced performance robustly across dataset size (Fig. 6c-d). Relative improvements were greater on larger datasets and longer rollouts. Additional results regarding network and dataset size, including spectra, are shown in fig. 26.

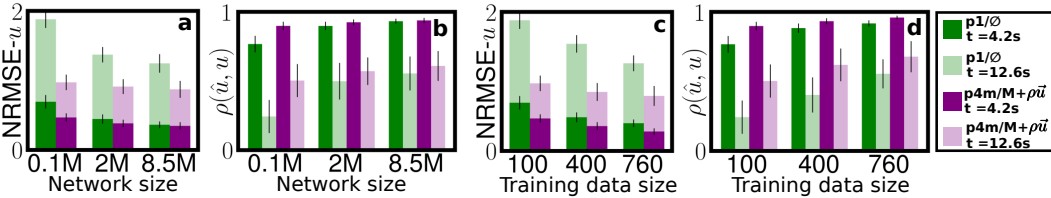

Figure 6: Accuracy of symmetry- and physics-constrained INS models across data and network sizes, at $t$ =4.2 s and 12.6 s. . (a-b) NRMSE-$u$ and $\rho(\hat{u}, u)$ vs. network size for p1/∅ and p4m/M+$\rho\vec{u}$. (c-d) NRMSE-$u$ and $\rho(\hat{u}, u)$ for p1/∅ and p4m/M+$\rho\vec{u}$ vs. training datasets size.

## 6 DISCUSSION

We enforced hard constraints on symmetries and conservation laws for neural PDE surrogates. We extended the applicability of previous techniques to c-grids, and systematically tested performance across tasks and constraints. Symmetries were more effective, but conservation laws were not redundant. Double constraints best matched reference simulations, individually and statistically.

**Limitations & Future work** For large enough networks and datasets, constraints might be learned from data (Stachenfeld et al., 2021; Watt-Meyer et al., 2023), but our results show the gap between constrained and unconstrained surrogates grows with rollout length even for large networks and datasets. Thus, constraints are likely relevant for longer time scales, e.g. for seasonal forecasts and climate projections (Kochkov et al., 2024; Watt-Meyer et al., 2023; Nguyen et al., 2023).

Our understanding of how these constraints limit error accumulation remains rudimentary. While we lack a rigorous theory, empirical investigations of how error accumulation correlates with constraint violations over time and ICs could provide some clarity. It also remains unclear how these improvements might transfer to other PDE types, such as hyperbolic equations.

We considered mass and momentum conservation, and symmetries of square 2D grids. Future work could pursue energy conservation (Cranmer et al., 2020), continuous symmetry groups (Cohen et al., 2018; Esteves et al., 2018), alternative grids and meshes (Cohen et al., 2019; De Haan et al., 2020), and other architectures and techniques, such as dilated Resnets, unrolled training, invariant measure learning, transformers and denoising diffusions (Takamoto et al., 2022; Brandstetter et al., 2022b; Schiff et al., 2024; List et al., 2024; Li et al., 2020; Lippe et al., 2024). Nonetheless, we believe that our results clearly demonstrate the potential of these inductive biases in improving rollout accuracy and extensiib to longer time scales.

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

# A    THE SYMMETRIES OF SWEs AND INE ON C-GRID STAGGERING

## A.1    GRID DISCRETIZATIONS

Here we show C-grid staggering for SWEs and INS.

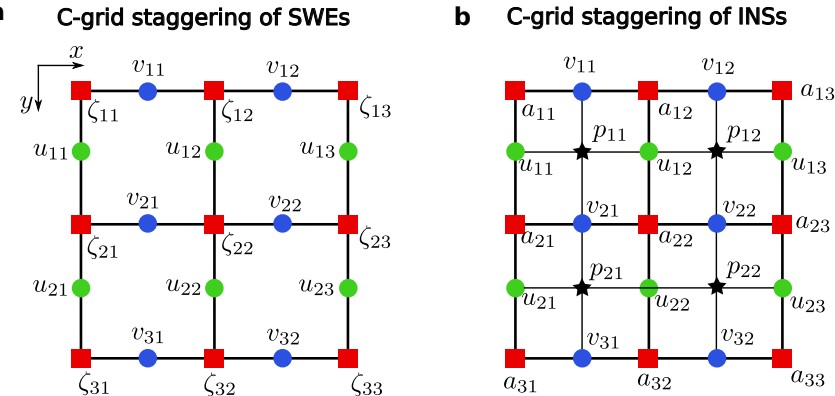

Figure 7: C-grid staggering for SWEs and INS.

The symmetry transformation of the numerical solver $S$ for SWEs can be described as:

$$\textbf{flip}: \quad S_\zeta(F(\zeta), F(u), -F(v)) = F(S_\zeta(\zeta, u, v)) \tag{10}$$

$$\textbf{rotation}: \quad S_\zeta(R(\zeta), -R(v), R(u)) = R(S_\zeta(\zeta, u, v)) \tag{11}$$

$$\textbf{flip} - \textbf{rotation}: \quad S_\zeta(R(F(\zeta)), R(F(v)), R(F(u))) = R(F(S_\zeta(\zeta, u, v))) \tag{12}$$

where $S_\zeta$ denotes numerically solving for $\zeta$ in the next time step, $F$ is a flipping operator, and $R$ is rotation.

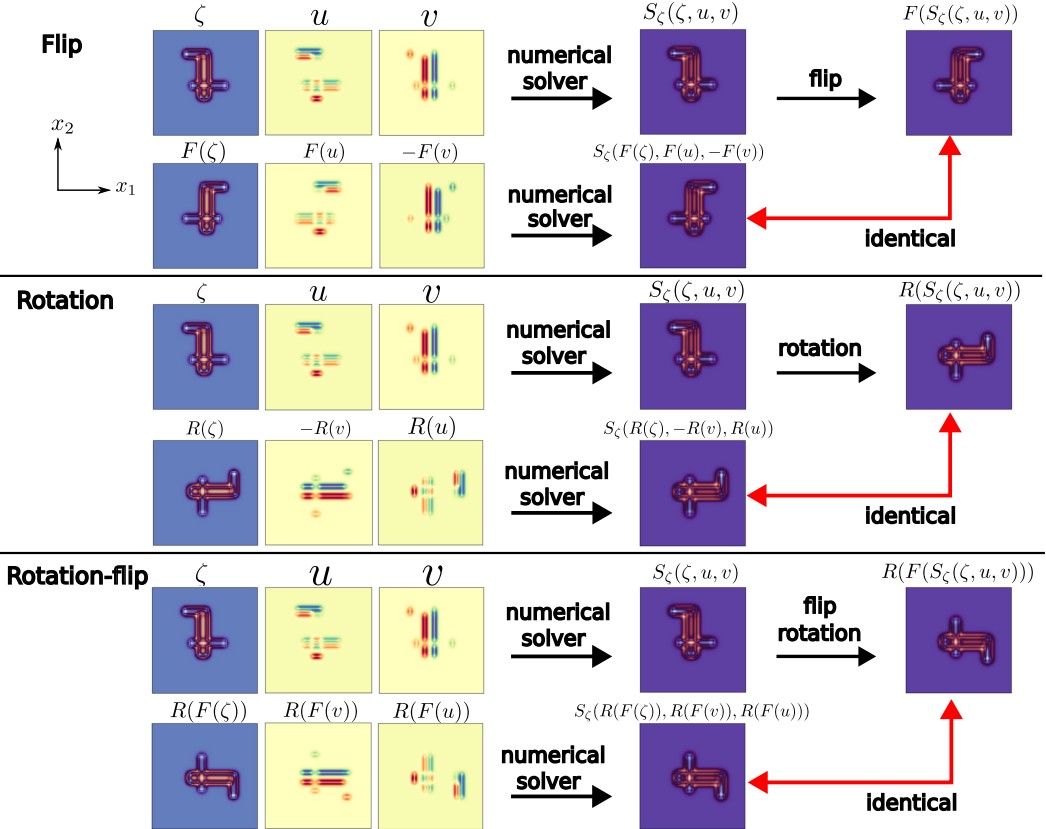

Figure 8: Empirical validation of the symmetries of the numerical SWE solver. Three transformations, flip, rotation, and flip-rotation are shown. These plots correspond to Eqs. 10-12

Next, the flip, rotation, and flip-rotation symmetries of of INSs can be described as follows:

$$\textbf{flip}: \begin{cases} S_u(-F(u), F(v)) = -F(S_u(u,v)) \\ S_v(-F(u), F(v)) = F(S_v(u,v)) \end{cases} \tag{13}$$

$$\textbf{rotation}: \begin{cases} S_u(R(v), -R(u)) = R(S_v(u,v)) \\ S_v(R(v), -R(u)) = R(-S_u(u,v)) \end{cases} \tag{14}$$

$$\textbf{flip}-\textbf{rotation}: \begin{cases} S_u(R(F(v)), -R(-F(u))) = R(F(S_v(u,v))) \\ S_v(R(F(v)), -R(-F(u))) = -R(-F(S_u(u,v))) \end{cases} \tag{15}$$

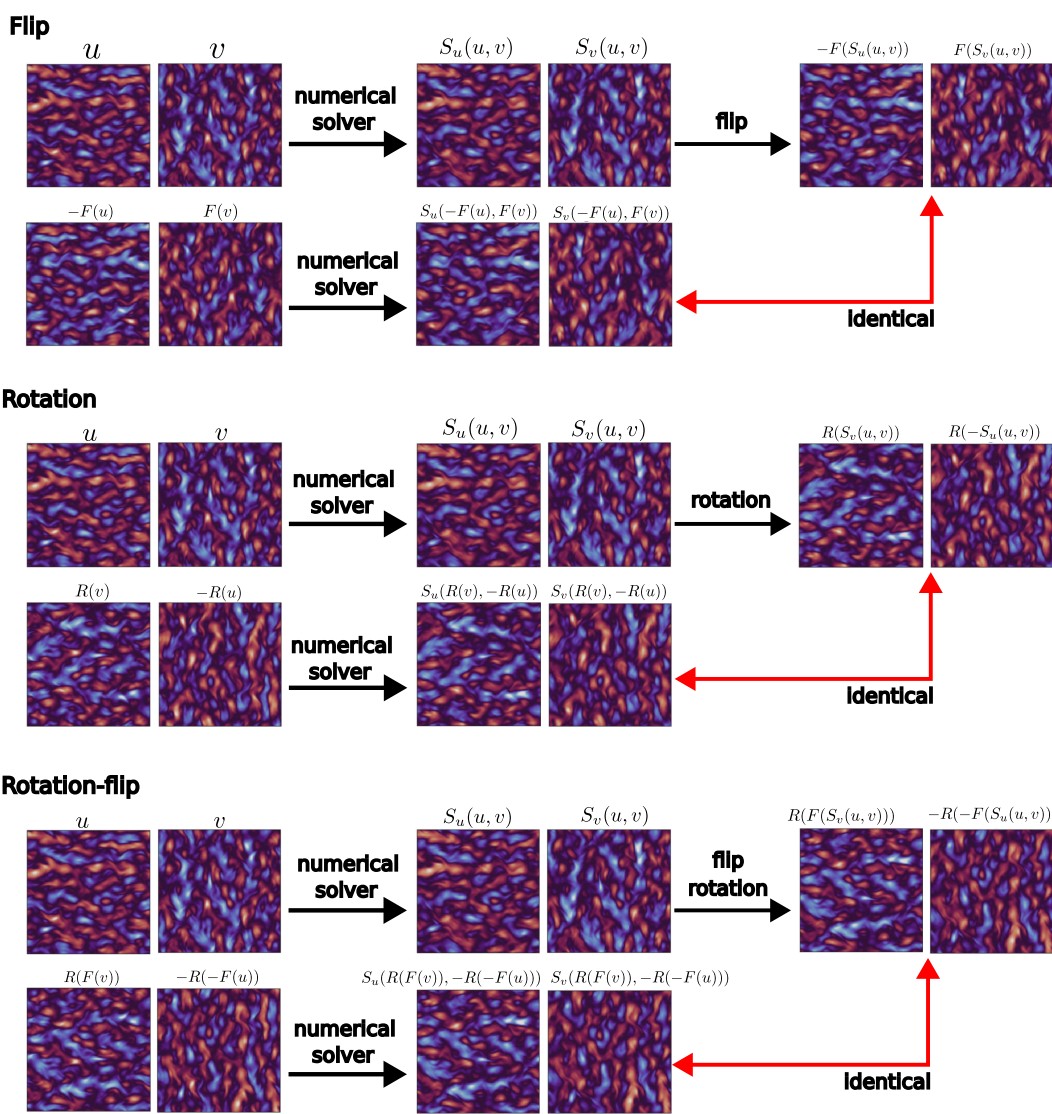

Figure 9: Empirical validation of the symmetries of the numerical INS solver. We show the symmetries of flip, rotation, flip-rotation for INS.

## B    PADDING OPTIONS

In some numerical solvers, although a C-grid staggering is used, the software produces output of the same size for each component of the vector field, requiring special attention to the chosen conventions for padding and boundary representation in the outputs. In this special case, a padding

technique is required to restore the vector field to the standard C-grid staggering. For example, periodic padding can be applied for the periodic BCs. It is important to note that the physical properties of the data, such as the divergence in the incompressible Navier–Stokes equations and the BCs, must remain unchanged when applying this correction.

## C GROUP EQUIVARIANT INPUT LAYERS

In this section, we write out explicit formulas for the equivariant input layers, and prove that they satisfy equivariance. For brevity we include proofs only for p4, but extension to p4m is trivial.

### C.1 GROUP EQUIVARIANT INPUT LAYER FOR SWES

Since our input data for shallow water equations (SWEs) uses C-grid staggering, as illustrated in Figure 7, we need to construct an input layer that matches the C-grid staggering while maintaining equivariance. On the C-grid, the variables $u$ and $v$ have different sizes. Therefore, we select two rectangular filters, $W^u_{j,i,\cdot,\cdot}$ and $W^v_{j,i,\cdot,\cdot}$, for $u$ and $v$, where the $\cdot$ symbol denotes all values along a given axis. The filter $W$ is an $c_{\text{in}} \times c_0 \times K \times S$ array, where $c_{\text{in}}$ is the number of input channels, $c_0$ is the batch size, and $K \times S$ represents the filter size. For instance, the sizes are $4 \times 3$ for $u$ and $3 \times 4$ for $v$. When performing group transformations in the input layer, we need to swap the filters for $u$ and $v$ to match the sizes of the input variables.

We first show an input layer of group p4 transformation which has four channels obtained from 4 different rotated filters. The detail input layer is written as following 4 equations:

$$y^1_{j,0,\cdot,\cdot} = \sum_{i=0}^{c^\zeta_{in}-1} \left( W^\zeta_{j,i,\cdot,\cdot} \star \zeta_{i,\cdot,\cdot} \right) + \sum_{i=0}^{c^u_{in}-1} \left( W^u_{j,i,\cdot,\cdot} \star u_{i,\cdot,\cdot} \right) + \sum_{i=0}^{c^v_{in}-1} \left( W^v_{j,i,\cdot,\cdot} \star v_{i,\cdot,\cdot} \right) + b_j, \quad (16)$$

$$y^1_{j,1,\cdot,\cdot} = \sum_{i=0}^{c^\zeta_{in}-1} \left( R^{90°}_{\text{rot}}(W^\zeta_{j,i,\cdot,\cdot}) \star \zeta_{i,\cdot,\cdot} \right) + \sum_{i=0}^{c^u_{in}-1} \left( -R^{90°}_{\text{rot}}(W^v_{j,i,\cdot,\cdot}) \star u_{i,\cdot,\cdot} \right)$$
$$+ \sum_{i=0}^{c^v_{in}-1} \left( R^{90°}_{\text{rot}}(W^u_{j,i,\cdot,\cdot}) \star v_{i,\cdot,\cdot} \right) + b_j, \quad (17)$$

$$y^1_{j,2,\cdot,\cdot} = \sum_{i=0}^{c^\zeta_{in}-1} \left( R^{180°}_{\text{rot}}(W^\zeta_{j,i,\cdot,\cdot}) \star \zeta_{i,\cdot,\cdot} \right) + \sum_{i=0}^{c^u_{in}-1} \left( -R^{180°}_{\text{rot}}(W^u_{j,i,\cdot,\cdot}) \star u_{i,\cdot,\cdot} \right)$$
$$+ \sum_{i=0}^{c^v_{in}-1} \left( -R^{180°}_{\text{rot}}(W^v_{j,i,\cdot,\cdot}) \star v_{i,\cdot,\cdot} \right) + b_j, \quad (18)$$

$$y^1_{j,3,\cdot,\cdot} = \sum_{i=0}^{c^\zeta_{in}-1} \left( R^{270°}_{\text{rot}}(W^\zeta_{j,i,\cdot,\cdot}) \star \zeta_{i,\cdot,\cdot} \right) + \sum_{i=0}^{c^u_{in}-1} \left( R^{270°}_{\text{rot}}(W^v_{j,i,\cdot,\cdot}) \star u_{i,\cdot,\cdot} \right)$$
$$+ \sum_{i=0}^{c^v_{in}-1} \left( -R^{270°}_{\text{rot}}(W^u_{j,i,\cdot,\cdot}) \star v_{i,\cdot,\cdot} \right) + b_j. \quad (19)$$

where $W^\zeta_{j,i,\cdot,\cdot}$ is a filter for $\zeta$ and it has a square size, for example, $4 \times 4$. $b$ is a $c_{\text{out}}$-element vector.

Next, we show an input layer for p4m using the same logic as the p4 input layer. It has 8 different group transformations including flip and rotation. The detail input layer is written as following equations:

$$y^1_{j,0,\cdot,\cdot} = \sum_{i=0}^{c^\zeta_{in}-1} \left( W^\zeta_{j,i,\cdot,\cdot} \star \zeta_{i,\cdot,\cdot} \right) + \sum_{i=0}^{c^u_{in}-1} \left( W^u_{j,i,\cdot,\cdot} \star u_{i,\cdot,\cdot} \right) + \sum_{i=0}^{c^v_{in}-1} \left( W^v_{j,i,\cdot,\cdot} \star v_{i,\cdot,\cdot} \right) + b_j, \quad (20)$$

$$y^1_{j,1,\cdot,\cdot} = \sum_{i=0}^{c^\zeta_{in}-1} \left( F_{\text{flip}}(W^\zeta_{j,i,\cdot,\cdot}) \star \zeta_{i,\cdot,\cdot} \right) + \sum_{i=0}^{c^u_{in}-1} \left( F_{\text{flip}}(W^u_{j,i,\cdot,\cdot}) \star u_{i,\cdot,\cdot} \right)$$
$$+ \sum_{i=0}^{c^v_{in}-1} \left( F_{\text{flip}}(W^v_{j,i,\cdot,\cdot}) \star v_{i,\cdot,\cdot} \right) + b_j, \quad (21)$$

$$y^1_{j,2,\cdot,\cdot} = \sum_{i=0}^{c^\zeta_{in}-1} \left( R^{90°}_{\text{rot}}(W^\zeta_{j,i,\cdot,\cdot}) \star \zeta_{i,\cdot,\cdot} \right) + \sum_{i=0}^{c^u_{in}-1} \left( R^{90°}_{\text{rot}}(W^v_{j,i,\cdot,\cdot}) \star u_{i,\cdot,\cdot} \right)$$
$$+ \sum_{i=0}^{c^v_{in}-1} \left( R^{90°}_{\text{rot}}(W^u_{j,i,\cdot,\cdot}) \star v_{i,\cdot,\cdot} \right) + b_j, \quad (22)$$

$$y^1_{j,3,\cdot,\cdot} = \sum_{i=0}^{c^\zeta_{in}-1} \left( F_{\text{flip}}(R^{90°}_{\text{rot}}(W^\zeta_{j,i,\cdot,\cdot})) \star \zeta_{i,\cdot,\cdot} \right) + \sum_{i=0}^{c^u_{in}-1} \left( F_{\text{flip}}(R^{90°}_{\text{rot}}(W^v_{j,i,\cdot,\cdot})) \star u_{i,\cdot,\cdot} \right)$$
$$+ \sum_{i=0}^{c^v_{in}-1} \left( F_{\text{flip}}(R^{90°}_{\text{rot}}(W^u_{j,i,\cdot,\cdot})) \star v_{i,\cdot,\cdot} \right) + b_j, \quad (23)$$

$$y^1_{j,4,\cdot,\cdot} = \sum_{i=0}^{c^\zeta_{in}-1} \left( R^{180°}_{\text{rot}}(W^\zeta_{j,i,\cdot,\cdot}) \star \zeta_{i,\cdot,\cdot} \right) + \sum_{i=0}^{c^u_{in}-1} \left( R^{180°}_{\text{rot}}(W^u_{j,i,\cdot,\cdot}) \star u_{i,\cdot,\cdot} \right)$$
$$+ \sum_{i=0}^{c^v_{in}-1} \left( R^{180°}_{\text{rot}}(W^v_{j,i,\cdot,\cdot}) \star v_{i,\cdot,\cdot} \right) + b_j, \quad (24)$$

$$y^1_{j,5,\cdot,\cdot} = \sum_{i=0}^{c^\zeta_{in}-1} \left( F_{\text{flip}}(R^{180°}_{\text{rot}}(W^\zeta_{j,i,\cdot,\cdot})) \star \zeta_{i,\cdot,\cdot} \right) + \sum_{i=0}^{c^u_{in}-1} \left( F_{\text{flip}}(R^{180°}_{\text{rot}}(W^u_{j,i,\cdot,\cdot})) \star u_{i,\cdot,\cdot} \right)$$
$$+ \sum_{i=0}^{c^v_{in}-1} \left( F_{\text{flip}}(R^{180°}_{\text{rot}}(W^v_{j,i,\cdot,\cdot})) \star v_{i,\cdot,\cdot} \right) + b_j, \quad (25)$$

$$y^1_{j,6,\cdot,\cdot} = \sum_{i=0}^{c^\zeta_{in}-1} \left( R^{270°}_{\text{rot}}(W^\zeta_{j,i,\cdot,\cdot}) \star \zeta_{i,\cdot,\cdot} \right) + \sum_{i=0}^{c^u_{in}-1} \left( R^{270°}_{\text{rot}}(W^v_{j,i,\cdot,\cdot}) \star u_{i,\cdot,\cdot} \right)$$
$$+ \sum_{i=0}^{c^v_{in}-1} \left( R^{270°}_{\text{rot}}(W^u_{j,i,\cdot,\cdot}) \star v_{i,\cdot,\cdot} \right) + b_j, \quad (26)$$

$$y^1_{j,7,\cdot,\cdot} = \sum_{i=0}^{c^\zeta_{in}-1} \left( F_{\text{flip}}(R^{270°}_{\text{rot}}(W^\zeta_{j,i,\cdot,\cdot})) \star \zeta_{i,\cdot,\cdot} \right) + \sum_{i=0}^{c^u_{in}-1} \left( F_{\text{flip}}(R^{270°}_{\text{rot}}(W^v_{j,i,\cdot,\cdot})) \star u_{i,\cdot,\cdot} \right)$$
$$+ \sum_{i=0}^{c^v_{in}-1} \left( F_{\text{flip}}(R^{270°}_{\text{rot}}(W^u_{j,i,\cdot,\cdot})) \star v_{i,\cdot,\cdot} \right) + b_j, \quad (27)$$

where the filters $W^u_{j,i,\cdot,\cdot}$ and $W^v_{j,i,\cdot,\cdot}$ are rectangles and the filter $W^\zeta_{j,i,\cdot,\cdot}$ is a square.

## C.2 GROUP EQUIVARIANCE OF P4 INPUT LAYER IN SWEs

Here we prove equivariance to the rotation symmetry of SWEs $S_\zeta(R(\zeta), -R(v), R(u)) = R(S_\zeta(\zeta, u, v))$. To prove the equivariance of our p4 input layer, we need to transform our input from $\zeta_{i,\cdot,\cdot}, u_{i,\cdot,\cdot}, v_{i,\cdot,\cdot}$ to $R_{\text{rot}}^{90°}(\zeta_{i,\cdot,\cdot}), -R_{\text{rot}}^{90°}(v_{i,\cdot,\cdot})$, and $R_{\text{rot}}^{90°}(u_{i,\cdot,\cdot})$. Then, using the p4 input layer shown in Equations (16-19), the transformed four layers $\tilde{y}$ become:

$$
\tilde{y}_{j,0,\cdot,\cdot}^1 = \sum_{i=0}^{c_{in}^\zeta-1} \left( W_{j,i,\cdot,\cdot}^\zeta \star R_{\text{rot}}^{90°}(\zeta_{i,\cdot,\cdot}) \right) + \sum_{i=0}^{c_{in}^u-1} \left( W_{j,i,\cdot,\cdot}^u \star -R_{\text{rot}}^{90°}(v_{i,\cdot,\cdot}) \right)
$$

$$
+ \sum_{i=0}^{c_{in}^v-1} \left( W_{j,i,\cdot,\cdot}^v \star R_{\text{rot}}^{90°}(u_{i,\cdot,\cdot}) \right) + b_j, \tag{28}
$$

$$
\tilde{y}_{j,1,\cdot,\cdot}^1 = \sum_{i=0}^{c_{in}^\zeta-1} \left( R_{\text{rot}}^{90°}(W_{j,i,\cdot,\cdot}^\zeta) \star R_{\text{rot}}^{90°}(\zeta_{i,\cdot,\cdot}) \right) + \sum_{i=0}^{c_{in}^u-1} \left( -R_{\text{rot}}^{90°}(W_{j,i,\cdot,\cdot}^v) \star -R_{\text{rot}}^{90°}(v_{i,\cdot,\cdot}) \right)
$$

$$
+ \sum_{i=0}^{c_{in}^v-1} \left( R_{\text{rot}}^{90°}(W_{j,i,\cdot,\cdot}^u) \star R_{\text{rot}}^{90°}(u_{i,\cdot,\cdot}) \right) + b_j, \tag{29}
$$

$$
\tilde{y}_{j,2,\cdot,\cdot}^1 = \sum_{i=0}^{c_{in}^\zeta-1} \left( R_{\text{rot}}^{180°}(W_{j,i,\cdot,\cdot}^\zeta) \star R_{\text{rot}}^{90°}(\zeta_{i,\cdot,\cdot}) \right) + \sum_{i=0}^{c_{in}^u-1} \left( -R_{\text{rot}}^{180°}(W_{j,i,\cdot,\cdot}^u) \star -R_{\text{rot}}^{90°}(v_{i,\cdot,\cdot}) \right)
$$

$$
+ \sum_{i=0}^{c_{in}^v-1} \left( -R_{\text{rot}}^{180°}(W_{j,i,\cdot,\cdot}^v) \star R_{\text{rot}}^{90°}(u_{i,\cdot,\cdot}) \right) + b_j, \tag{30}
$$

$$
\tilde{y}_{j,3,\cdot,\cdot}^1 = \sum_{i=0}^{c_{in}^\zeta-1} \left( R_{\text{rot}}^{270°}(W_{j,i,\cdot,\cdot}^\zeta) \star R_{\text{rot}}^{90°}(\zeta_{i,\cdot,\cdot}) \right) + \sum_{i=0}^{c_{in}^u-1} \left( R_{\text{rot}}^{270°}(W_{j,i,\cdot,\cdot}^v) - R_{\text{rot}}^{90°}(v_{i,\cdot,\cdot}) \right)
$$

$$
+ \sum_{i=0}^{c_{in}^v-1} \left( -R_{\text{rot}}^{270°}(W_{j,i,\cdot,\cdot}^u) \star R_{\text{rot}}^{90°}(u_{i,\cdot,\cdot}) \right) + b_j. \tag{31}
$$

Next, we need to rotate 90 degree for the first layer output in Equations (16)-(19) and then put rotation into the convolution. We obtain the following equations:

$$
\begin{aligned}
R_{\text{rot}}^{90°}(y_{j,0,\cdot,\cdot}^1) = R_{\text{rot}}^{90°}\Bigg( & \sum_{i=0}^{c_{in}^{\zeta}-1}\left(W_{j,i,\cdot,\cdot}^{\zeta}\star\zeta_{i,\cdot,\cdot}\right) + \sum_{i=0}^{c_{in}^{u}-1}\left(W_{j,i,\cdot,\cdot}^{u}\star u_{i,\cdot,\cdot}\right) \\
& + \sum_{i=0}^{c_{in}^{v}-1}\left(W_{j,i,\cdot,\cdot}^{v}\star v_{i,\cdot,\cdot}\right) + b_j\Bigg) = \tilde{y}_{j,1,\cdot,\cdot}^1.
\end{aligned}
\tag{32}
$$

$$
\begin{aligned}
R_{\text{rot}}^{90°}(y_{j,1,\cdot,\cdot}^1) = R_{\text{rot}}^{90°}\Bigg( & \sum_{i=0}^{c_{in}^{\zeta}-1}\left(R_{\text{rot}}^{90°}(W_{j,i,\cdot,\cdot}^{\zeta})\star\zeta_{i,\cdot,\cdot}\right) + \sum_{i=0}^{c_{in}^{u}-1}\left(-R_{\text{rot}}^{90°}(W_{j,i,\cdot,\cdot}^{v})\star u_{i,\cdot,\cdot}\right) \\
& + \sum_{i=0}^{c_{in}^{v}-1}\left(R_{\text{rot}}^{90°}(W_{j,i,\cdot,\cdot}^{u})\star v_{i,\cdot,\cdot}\right) + b_j\Bigg) = \tilde{y}_{j,2,\cdot,\cdot}^1.
\end{aligned}
\tag{33}
$$

$$
\begin{aligned}
R_{\text{rot}}^{90°}(y_{j,2,\cdot,\cdot}^1) = R_{\text{rot}}^{90°}\Bigg( & \sum_{i=0}^{c_{in}^{\zeta}-1}\left(R_{\text{rot}}^{180°}(W_{j,i,\cdot,\cdot}^{\zeta})\star\zeta_{i,\cdot,\cdot}\right) + \sum_{i=0}^{c_{in}^{u}-1}\left(-R_{\text{rot}}^{180°}(W_{j,i,\cdot,\cdot}^{u})\star u_{i,\cdot,\cdot}\right) \\
& + \sum_{i=0}^{c_{in}^{v}-1}\left(-R_{\text{rot}}^{180°}(W_{j,i,\cdot,\cdot}^{v})\star v_{i,\cdot,\cdot}\right) + b_j\Bigg) = \tilde{y}_{j,3,\cdot,\cdot}^1.
\end{aligned}
\tag{34}
$$

$$
\begin{aligned}
R_{\text{rot}}^{90°}(y_{j,3,\cdot,\cdot}^1) = R_{\text{rot}}^{90°}\Bigg( & \sum_{i=0}^{c_{in}^{\zeta}-1}\left(R_{\text{rot}}^{270°}(W_{j,i,\cdot,\cdot}^{\zeta})\star\zeta_{i,\cdot,\cdot}\right) + \sum_{i=0}^{c_{in}^{u}-1}\left(R_{\text{rot}}^{270°}(W_{j,i,\cdot,\cdot}^{v})\star u_{i,\cdot,\cdot}\right) \\
& + \sum_{i=0}^{c_{in}^{v}-1}\left(-R_{\text{rot}}^{270°}(W_{j,i,\cdot,\cdot}^{u})\star v_{i,\cdot,\cdot}\right) + b_j\Bigg) = \tilde{y}_{j,0,\cdot,\cdot}^1.
\end{aligned}
\tag{35}
$$

We find these equations satisfy the formula:

$$
R_{\text{rot}}^{90°}(y^1(\zeta_{i,\cdot,\cdot}, u_{i,\cdot,\cdot}, v_{i,\cdot,\cdot})) = \tilde{y}^1(R_{\text{rot}}^{90°}(\zeta_{i,\cdot,\cdot}), -R_{\text{rot}}^{90°}(v_{i,\cdot,\cdot}), R_{\text{rot}}^{90°}(u_{i,\cdot,\cdot}))
\tag{36}
$$

This form precisely matches the rotation symmetry for SWEs in Equation(11). Thus, we have proved the group equivariant of p4 input layer in shallow water equations.

### C.3 GROUP EQUIVARIANT INPUT LAYER FOR INS

The input data of incompressible Navier–Stokes equations is the velocity $u$ and $v$, which have different sizes on the C-grid staggering. Thus, we also need two rectangle filters $W_{j,i,\cdot,\cdot}^u$ and $W_{j,i,\cdot,\cdot}^v$ for the velocity field. According to the symmetries of rotation of INS in Equation (14), we first build a p4 input layer for INS as following equations:

$$
y_{j,0,\cdot,\cdot}^1 = \sum_{i=0}^{c_{in}^u-1}\left(W_{j,i,\cdot,\cdot}^u\star u_{i,\cdot,\cdot}\right) + \sum_{i=0}^{c_{in}^v-1}\left(W_{j,i,\cdot,\cdot}^v\star v_{i,\cdot,\cdot}\right) + b_j,
\tag{37}
$$

$$
y_{j,1,\cdot,\cdot}^1 = \sum_{i=0}^{c_{in}^u-1}\left(R_{\text{rot}}^{90°}(W_{j,i,\cdot,\cdot}^v)\star u_{i,\cdot,\cdot}\right) + \sum_{i=0}^{c_{in}^v-1}\left(-R_{\text{rot}}^{90°}(W_{j,i,\cdot,\cdot}^u)\star v_{i,\cdot,\cdot}\right) + b_j,
\tag{38}
$$

$$
y_{j,2,\cdot,\cdot}^1 = \sum_{i=0}^{c_{in}^u-1}\left(-R_{\text{rot}}^{180°}(W_{j,i,\cdot,\cdot}^u)\star u_{i,\cdot,\cdot}\right) + \sum_{i=0}^{c_{in}^v-1}\left(-R_{\text{rot}}^{180°}(W_{j,i,\cdot,\cdot}^v)\star v_{i,\cdot,\cdot}\right) + b_j,
\tag{39}
$$

$$
y_{j,3,\cdot,\cdot}^1 = \sum_{i=0}^{c_{in}^u-1}\left(-R_{\text{rot}}^{270°}(W_{j,i,\cdot,\cdot}^v)\star u_{i,\cdot,\cdot}\right) + \sum_{i=0}^{c_{in}^v-1}\left(R_{\text{rot}}^{270°}(W_{j,i,\cdot,\cdot}^u)\star v_{i,\cdot,\cdot}\right) + b_j.
\tag{40}
$$

Next, based on the symmetries of flip-rotation of INS in Equation (15), we introduce a p4m of input layer as following equations:

$$y_{j,0,\cdot,\cdot}^{1} = \sum_{i=0}^{c_{in}^{u}-1} \left( W_{j,i,\cdot,\cdot}^{u} \star u_{i,\cdot,\cdot} \right) + \sum_{i=0}^{c_{in}^{v}-1} \left( W_{j,i,\cdot,\cdot}^{v} \star v_{i,\cdot,\cdot} \right) + b_j, \tag{41}$$

$$y_{j,1,\cdot,\cdot}^{1} = \sum_{i=0}^{c_{in}^{u}-1} \left( F_{\text{flip}}(W_{j,i,\cdot,\cdot}^{u}) \star u_{i,\cdot,\cdot} \right) + \sum_{i=0}^{c_{in}^{v}-1} \left( F_{\text{flip}}(W_{j,i,\cdot,\cdot}^{v}) \star v_{i,\cdot,\cdot} \right) + b_j, \tag{42}$$

$$y_{j,2,\cdot,\cdot}^{1} = \sum_{i=0}^{c_{in}^{u}-1} \left( R_{\text{rot}}^{90^\circ}(W_{j,i,\cdot,\cdot}^{v}) \star u_{i,\cdot,\cdot} \right) + \sum_{i=0}^{c_{in}^{v}-1} \left( R_{\text{rot}}^{90^\circ}(W_{j,i,\cdot,\cdot}^{u}) \star v_{i,\cdot,\cdot} \right) + b_j, \tag{43}$$

$$y_{j,3,\cdot,\cdot}^{1} = \sum_{i=0}^{c_{in}^{u}-1} \left( F_{\text{flip}}(R_{\text{rot}}^{90^\circ}(W_{j,i,\cdot,\cdot}^{v})) \star u_{i,\cdot,\cdot} \right) + \sum_{i=0}^{c_{in}^{v}-1} \left( F_{\text{flip}}(R_{\text{rot}}^{90^\circ}(W_{j,i,\cdot,\cdot}^{u})) \star v_{i,\cdot,\cdot} \right) + b_j, \tag{44}$$

$$y_{j,4,\cdot,\cdot}^{1} = \sum_{i=0}^{c_{in}^{u}-1} \left( R_{\text{rot}}^{180^\circ}(W_{j,i,\cdot,\cdot}^{u}) \star u_{i,\cdot,\cdot} \right) + \sum_{i=0}^{c_{in}^{v}-1} \left( R_{\text{rot}}^{180^\circ}(W_{j,i,\cdot,\cdot}^{v}) \star v_{i,\cdot,\cdot} \right) + b_j, \tag{45}$$

$$y_{j,5,\cdot,\cdot}^{1} = \sum_{i=0}^{c_{in}^{u}-1} \left( F_{\text{flip}}(R_{\text{rot}}^{180^\circ}(W_{j,i,\cdot,\cdot}^{u})) \star u_{i,\cdot,\cdot} \right) + \sum_{i=0}^{c_{in}^{v}-1} \left( F_{\text{flip}}(R_{\text{rot}}^{180^\circ}(W_{j,i,\cdot,\cdot}^{v})) \star v_{i,\cdot,\cdot} \right) + b_j, \tag{46}$$

$$y_{j,6,\cdot,\cdot}^{1} = \sum_{i=0}^{c_{in}^{u}-1} \left( R_{\text{rot}}^{270^\circ}(W_{j,i,\cdot,\cdot}^{v}) \star u_{i,\cdot,\cdot} \right) + \sum_{i=0}^{c_{in}^{v}-1} \left( R_{\text{rot}}^{270^\circ}(W_{j,i,\cdot,\cdot}^{u}) \star v_{i,\cdot,\cdot} \right) + b_j, \tag{47}$$

$$y_{j,7,\cdot,\cdot}^{1} = \sum_{i=0}^{c_{in}^{u}-1} \left( F_{\text{flip}}(R_{\text{rot}}^{270^\circ}(W_{j,i,\cdot,\cdot}^{v})) \star u_{i,\cdot,\cdot} \right) + \sum_{i=0}^{c_{in}^{v}-1} \left( F_{\text{flip}}(R_{\text{rot}}^{270^\circ}(W_{j,i,\cdot,\cdot}^{u})) \star v_{i,\cdot,\cdot} \right) + b_j, \tag{48}$$

## C.4 EMPIRICAL VALIDATION OF EQUIVARIANCE FOR SWE AND INS INPUT LAYERS

In this section, we plot an example showing the action of group equivariant input layers for SWEs and INS. First, we show the plot p4 and p4m group equivariant input layer of SWEs in fig. 10.

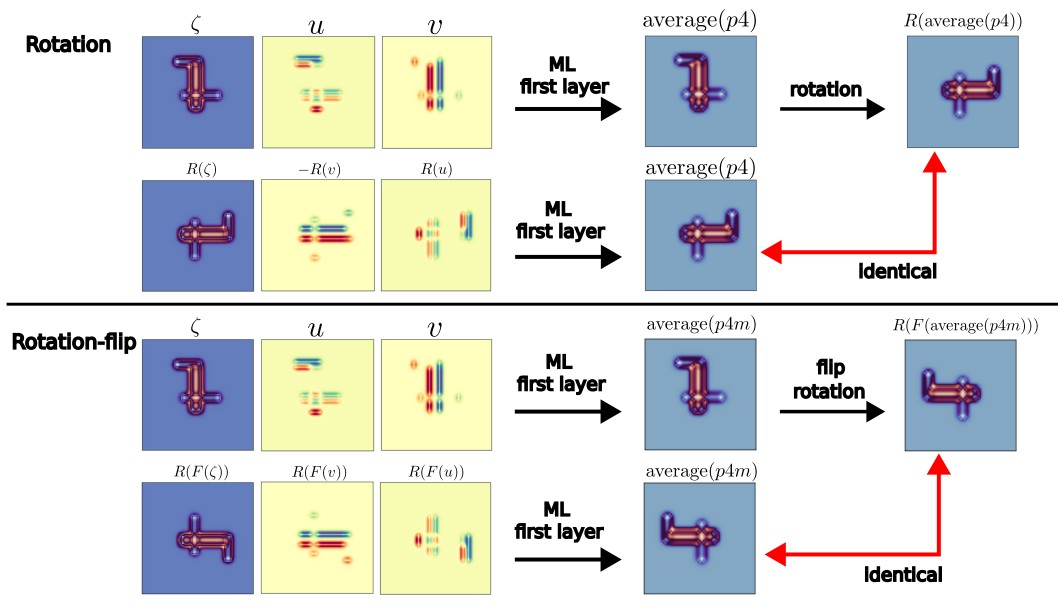

Figure 10: An example plot of input layer for the group p4 and p4m equivariant in shallow water equations. It clearly shows that our input layers are equivariant.

Next, we plot an example of p4 and p4m group equivariant for INS in fig. 11. We can find that our p4 and p4m input layers are equivariant.

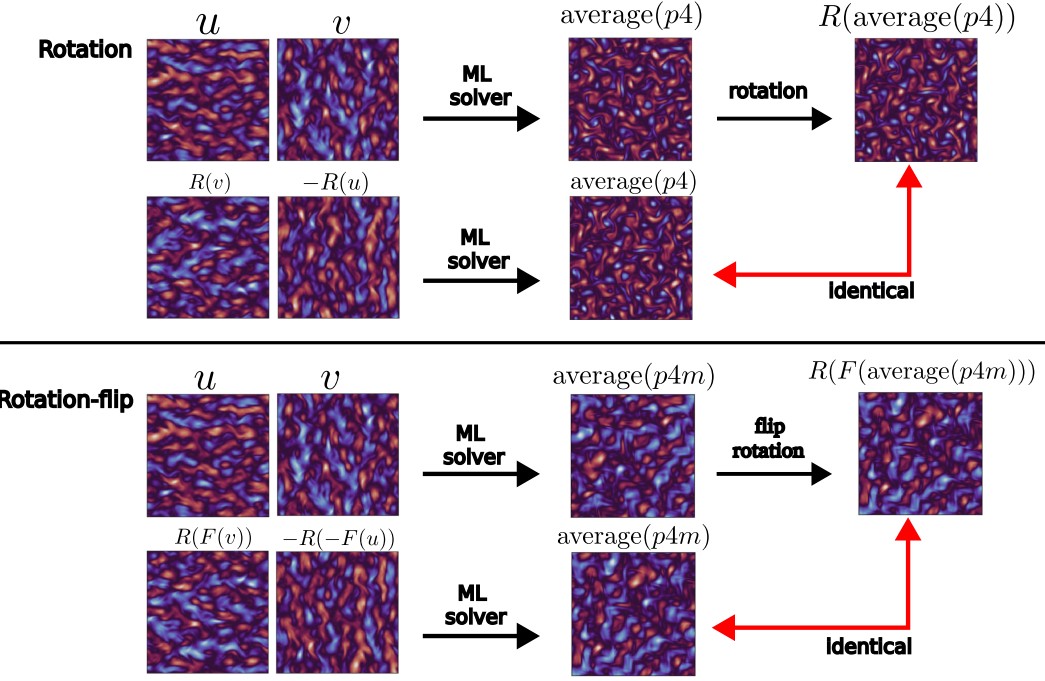

Figure 11: The group equivariant plots for the p4 and p4m input layers in incompressible Navier–Stokes equations.

## D  GROUP EQUIVARIANT OUTPUT VECTOR LAYER WITH C-GRID STAGGERING

After the modern U-Net, we need to select a feature field for the output based on the physical variables. For a scalar field, we can directly use `r2_act.trivial_repr` in escnn. However, for a vector field on C-grid staggering, we cannot use the vector field in escnn, referred to as `r2_act.irrep(1)`, because it is not on the C-grid and does not satisfy the symmetry of the discretized PDEs. Therefore, we build vector layers for p4 and p4m:

$$\mathbf{p4}: \; u_{i+0.5,j} = p_{i+1,j,0} - p_{i,j,1}; \quad v_{i,j+0.5} = p_{i,j+1,2} - p_{i,j,3} \tag{49}$$

$$\mathbf{p4m}: \begin{cases} u_{i+0.5,j} = p_{i+1,j,1} - p_{i,j,3} + p_{i+1,j,5} - p_{i,j,7} \\ v_{i,j+0.5} = p_{i,j+1,2} - p_{i,j,4} + p_{i,j+1,6} - p_{i,j,0} \end{cases} \tag{50}$$

where $p_{i,j,k}$ is on the regular representations. $i$ and $j$ express the position index and $k$ is the group transformation. $i + 0.5$ and $j + 0.5$ are position on C-grids for $u$ and $v$. Thus, these layers not only satisfy the group equivariant but also back to C-grid. An example of vector layer for P4 is shown in Figure (1) as red box.

### D.1  PROOF OF EQUIVARIANCE BY CONSTRUCTION FOR VECTOR OUTPUT LAYERS

In Equations (49-50), we show the vector output layers from p4 and p4m regular representation. Here, we show a process how we these layers were derived from the constraints we wish to prove.

**Output vector from p4 regular representation** First, we give a proof of the equivariance for our vector output layer for p4. For given input $u$ and $v$, in the regular representation layer, we have four channels related to p4 group transformation $p_{i+1,j,0}$, $p_{i+1,j,1}$, $p_{i+1,j,2}$, and $p_{i+1,j,3}$. When we

transform the input to $R(v), -R(u)$, the four regular representations become to $\hat{p}_{i+1,j,0}$, $\hat{p}_{i+1,j,1}$, $\hat{p}_{i+1,j,2}$ and $\hat{p}_{i+1,j,3}$. There exits a relation between them as:

$$R(p_{i+1,j,0}) = \hat{p}_{i+1,j,3} \tag{51}$$
$$R(p_{i+1,j,1}) = \hat{p}_{i+1,j,0} \tag{52}$$
$$R(p_{i+1,j,2}) = \hat{p}_{i+1,j,1} \tag{53}$$
$$R(p_{i+1,j,3}) = \hat{p}_{i+1,j,2} \tag{54}$$

This relation of p4 is also show in Figure(1) of hidden layers. Now we assume an equivariant output on c-grids staggering written as:

$$u_{i+0.5,j} = \sum_{k=0}^{3} c_k p_{i+1,j,k} - \sum_{k=0}^{3} d_k p_{i,j,k} \tag{55}$$

$$v_{i,j+0.5} = \sum_{k=0}^{3} e_k p_{i,j+1,k} - \sum_{k=0}^{3} f_k p_{i,j,k} \tag{56}$$

where $c_k$, $d_k$, $e_k$, and $f_k$ are coefficients. We can write the rotated output as:

$$\hat{u}_{i+0.5,j} = \sum_{k=0}^{3} c_k \hat{p}_{i+1,j,k} - \sum_{k=0}^{3} d_k \hat{p}_{i,j,k} \tag{57}$$

$$\hat{v}_{i,j+0.5} = \sum_{k=0}^{3} e_k \hat{p}_{i,j+1,k} - \sum_{k=0}^{3} f_k \hat{p}_{i,j,k} \tag{58}$$

According to the symmetry of p4 in Equation(14) for the vector field on c-grids, we can write $\hat{u}_{i+0.5,j} = R(v_{i,j+0.5})$; $\hat{v}_{i,j+0.5} = R(-u_{i+0.5,j})$. Combining all equations (51 -58) into the symmetry constraint. We obtain the relations for the coefficients:

$$c_1 = d_2 = e_3 = f_4 \tag{59}$$
$$c_2 = d_3 = e_4 = f_1 \tag{60}$$
$$c_3 = d_4 = e_1 = f_2 \tag{61}$$
$$c_4 = d_1 = e_2 = f_3 \tag{62}$$

We choose a simple case $c_1 = 1$ and $c_2 = c_3 = c_4 = 0$ in this work. Therefore, we obtain an equivariant vector output from p4 regular representation can be written as

$$u_{i+0.5,j} = p_{i+1,j,1} - p_{i,j,3} \tag{63}$$
$$v_{i,j+0.5} = p_{i,j+1,2} - p_{i,j,4} \tag{64}$$

**Output vector from p4m regular representation**  The p4m regular representation layer has 8 channels denoted as $p_{i+1,j,k}$ where $k = 0, \cdots, 7$. We also employ the same form as the p4 to build the vector output layer:

$$u_{i+0.5,j} = \sum_{k=0}^{7} c_k p_{i+1,j,k} - \sum_{k=0}^{7} d_k p_{i,j,k} \tag{65}$$

$$v_{i,j+0.5} = \sum_{k=0}^{7} e_k p_{i,j+1,k} - \sum_{k=0}^{7} f_k p_{i,j,k} \tag{66}$$

where the coefficients are $c_k$, $d_k$, $e_k$, $f_k$, where $k = 0, \cdots, 7$. Taking the same way of analysis like to the p4, we obtain the relation for each coefficient as following:

$$c_1 = d_2 = e_3 = f_4 = c_5 = d_6 = e_7 = f_0 \tag{67}$$
$$c_2 = d_3 = e_4 = f_1 = c_6 = d_7 = e_0 = f_5 \tag{68}$$
$$c_3 = d_4 = e_1 = f_2 = c_7 = d_0 = e_5 = f_6 \tag{69}$$
$$c_4 = d_1 = e_2 = f_3 = c_0 = d_5 = e_6 = f_7 \tag{70}$$

Here, we take a simple case $c_1 = 1$ and $c_2 = c_3 = c_4 = 0$. Thus, the vector output layers on c-grids for $u$ and $v$ from p4m regular representation are written as:

$$u_{i+0.5,j} = p_{i+1,j,1} - p_{i,j,3} + p_{i+1,j,5} - p_{i,j,7} \tag{71}$$
$$v_{i,j+0.5} = p_{i,j+1,2} - p_{i,j,4} + p_{i,j+1,6} - p_{i,j,0}. \tag{72}$$

# E  PHYSICAL CONSTRAINTS EMBEDDED INTO NETWORKS

For the SWE, the mass is a conserved variable. To enforce mass conservation during training and inference, we subtract the mean of the tendency update from iteself:

$$\zeta^{t+1} = \zeta^t + d\zeta - \text{mean}(d\zeta) \tag{73}$$

To conserve momentum for the INS at training/inference, we follow a similar approach to SWE training. We introduce another physical constraint by learning a scalar potential $a$ in Equation 6 using the neural network and update the velocity fields by taking the curl of that scalar field. Both constraints can be written as following:

$$\textbf{Momentum} - \textbf{conser.} : \ u^{t+1} = u^t + du - \text{mean}(du); \quad v^{t+1} = v^t + dv - \text{mean}(dv) \tag{74}$$

$$\textbf{Learn} - \textbf{scalar} - \textbf{potential} : \ u^{t+1} = u^t - \frac{\partial a}{\partial y}; \quad v^{t+1} = v^t + \frac{\partial a}{\partial x} \ . \tag{75}$$

These physical constraint layers are added following the output layers. An example implementation for INS can be found in the blue box in Figure (1).

Alternatively, we might have learned fluxes at the C-grid interfaces for conserved quantities at cell centers, or fluxes at the vertices for conserved quantities at the interfaces, similar to a finite volume solver. This would have the advantage of being locally computed, allowing easier generalization of domain size after training. We leave this avenue of exploration to future work, anticipating that further improvements in accuracy might be obtained.

## F Simulation parameters

In this section, we show the detail parameters used for solving the shallow water equations and incompressible Navier–Stokes equations for the data generation.

Table 3: Simulation parameters used for SWEs

| Parameters | Explanation | Value |
|---|---|---|
| $L \times L$ | simulation domain | $1000 \times 1000 \, (Km)$ |
| $d$ | undisturbed water depth | $100 \, (m)$ |
| $C_D$ | bottom drag coefficient | $1.0e-3$ |
| $g$ | acceleration due to gravity | $9.81 \, (m/s^2)$ |
| $\Delta x$ | space step | $10 \, (Km)$ |
| $\Delta t$ | time step | $300 \, (s)$ |
| $w_{\text{imp}}$ | implicit weighting | $0.5$ |

Table 4: Simulation parameters used for INS

| Parameters | Explanation | Value |
|---|---|---|
| $L \times L$ | simulation domain | $2\pi \times 2\pi$ |
| $\rho$ | density | $1$ |
| $\mu$ | viscosity | $1e-3$ |
| $T$ | simulation time | $224.34 \, s$ |
| $\Delta t_{\text{solver}}$ | the time step of numerical solver | $0.00436 \, s$ |
| $M \times M$ | the grids of numerical solver | $576 \times 576$ |
| $\Delta x_{\text{solver}}$ | the space step of numerical solver | $0.0109$ |
| $\Delta t_{\text{ml}}$ | the time step of ML model | $0.8375$ |
| $m \times m$ | the grids of ML model | $48 \times 48$ |
| $\Delta x_{\text{ml}}$ | the space step of ML solver | $0.1308$ |

## G A hybrid method for the prediction of shallow water system

Fig. 12 shows a hybrid method used to predict the solution of shallow water system. In our neural integrator, we only have one output $\zeta$ and we have three inputs $u$, $v$ and $\zeta$. Thus, we need a small numerical solver to calculate $u^t$ and $v^t$ from a given $\zeta^t$. These calculations are made only for autoregressive rollouts with trained networks, and not during training (Backhaus, 1983). The formulas for the velocity at the new time step can be written as

$$u^{n+1} = u^* - \Delta t g w_{\text{imp}} \frac{\partial \zeta^{n+1}}{\partial x} \tag{76}$$

$$v^{n+1} = v^* - \Delta t g w_{\text{imp}} \frac{\partial \zeta^{n+1}}{\partial y} \tag{77}$$

where $u^*$ and $v^*$ are written as

$$u^* = u^n - \Delta t c_D \frac{1}{h} u^n |u^n| - \Delta t g (1 - w_{\text{imp}}) \frac{\partial \zeta^n}{\partial x} + \Delta t a_h \nabla^2 u^n \tag{78}$$

$$v^* = v^n - \Delta t c_D \frac{1}{h} v^n |v^n| - \Delta t g (1 - w_{\text{imp}}) \frac{\partial \zeta^n}{\partial y} + \Delta t a_h \nabla^2 v^n \tag{79}$$

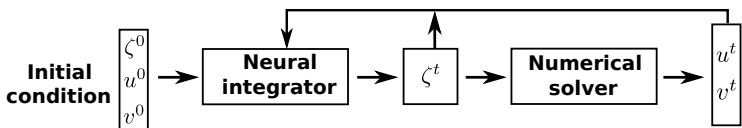

Figure 12: A structure of hybrid method for the prediction of shallow water system. Due to only one output $\zeta$, we need a small numerical solver to calculate $u^t$ and $v^t$.

# H    THE DETAILED RESULTS

## H.1    THE DETAILS FOR CLOSED BOUNDARY SHALLOW WATER SYSTEM

Figure 13 presents the predictions on the surface elevations $\zeta$ for additional time steps and models. This example is presented in Figure(3) of the main text.

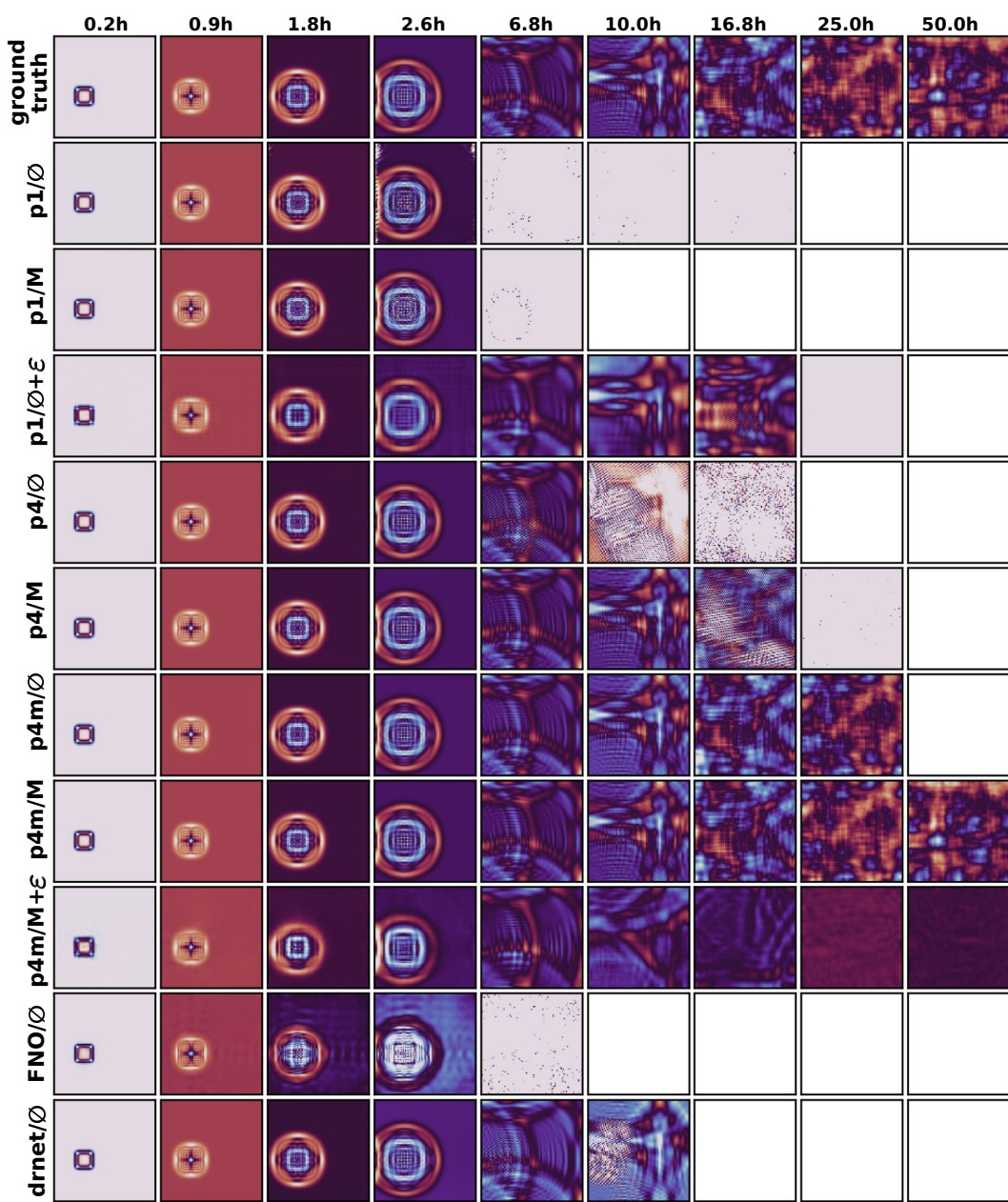

Figure 13: An example of rollout predictions on $\zeta$ from all methods for the SWE at different time steps. The top row shows the ground truth as a reference. It shows that p4m/M has the best long rollout accuracy.

We compare our best symmetry-physics-constrained model, p4m/M, with its noisy variant, where Gaussian noise with a zero mean and a standard deviation of 0.0001, $\mathcal{N}(\mu = 0, \sigma = 0.0001)$ is added during training. We find that training with input noise achieves long rollouts but with lower accuracy than the noise-free model. An example of the noisy approach's performance is shown in Figure (13). Predictions from the noisy model are less accurate, even at the early stages of the rollout.

Furthermore, the mass, momentum, and total energy for the shallow water equations are plotted over all tested methods over the course of a 50-hour period and presented in Figure 15. In the context of

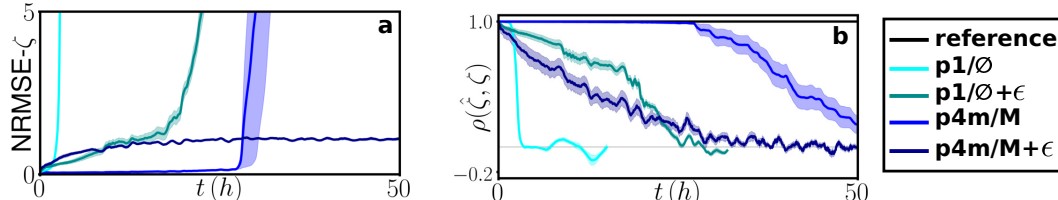

Figure 14: A comparison of predictions from methods that include noise during training, p1/∅ + ε and p4m/M+ε, with the no-noise approaches, p1/∅ and p4m/M, is presented using the metrics NRMSE-ζ and $\rho(\hat{\zeta}, \zeta)$. The NRMSE-ζ metric shows that p4m/M+ε maintains a relatively lower error over a longer time period compared to the other methods.

evolutionary processes, the mass, momentum, and total energy remain constant, as illustrated in the reference by the black curve. The p4m/M model demonstrates superior performance in maintaining mass conversion over extended periods. In terms of both momentum and total energy, the p4m/M model also demonstrates superior performance in comparison to all other models.

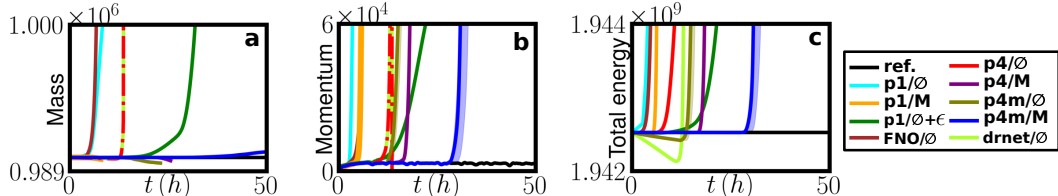

Figure 15: The mass, momentum, and total energy of the shallow water equations for all tested models over the course of 50 hours are presented in the Figure. Our most effective model, p4m/M, demonstrates superior performance in maintaining the conversion of mass, momentum, and energy in comparison to other methods.

## H.2 THE DETAILS FOR DECAYING TURBULENCE

Figure 16 and 17 presents the predictions on the velocity $u$ and $v$ over additional time steps and models,

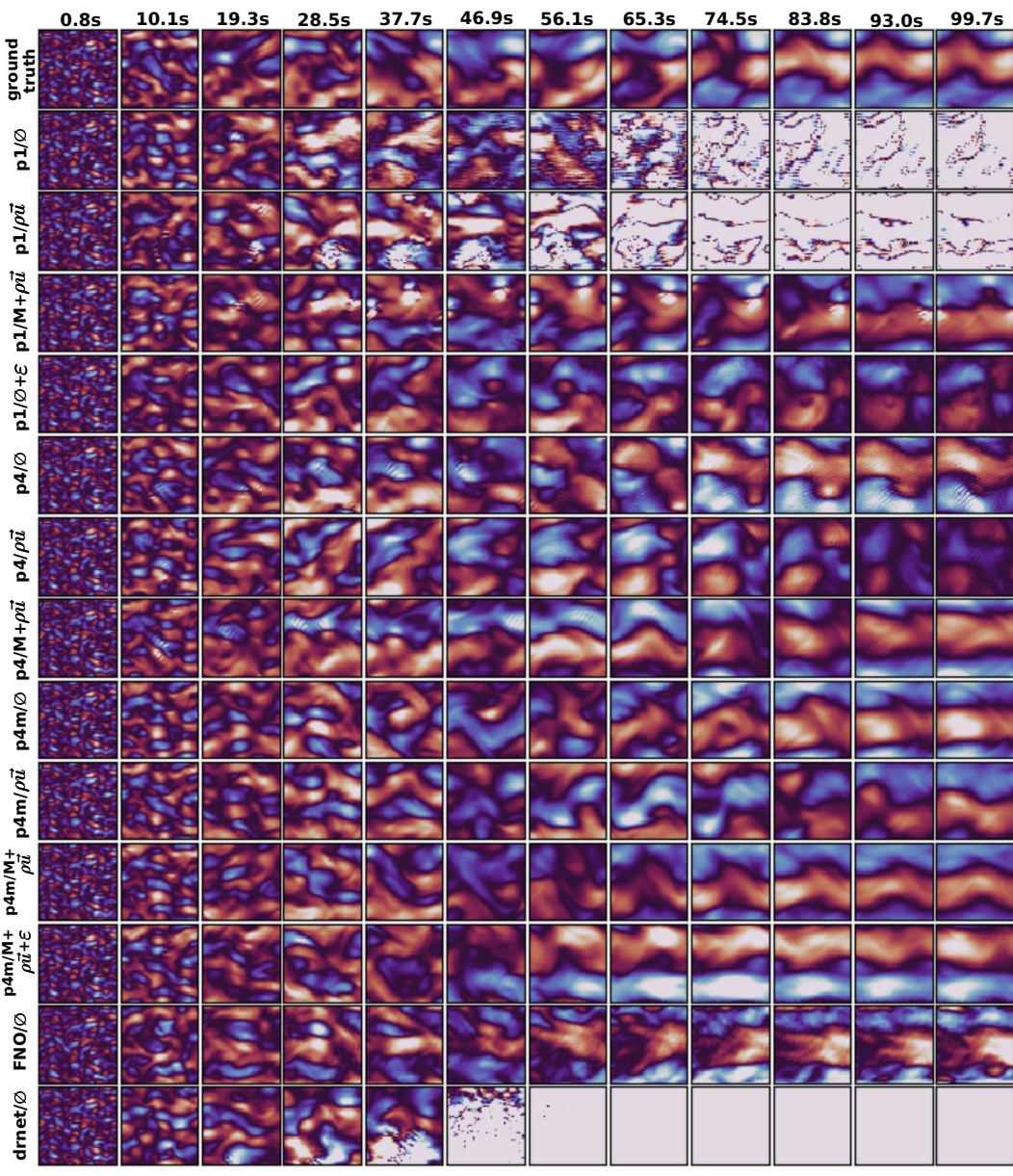

Figure 16: An example of rollout predictions on $u$ from all methods with network sizes of approximately 0.1M parameters for the incompressible Navier-Stokes equations at different time steps. The top row shows the ground truth as a reference.

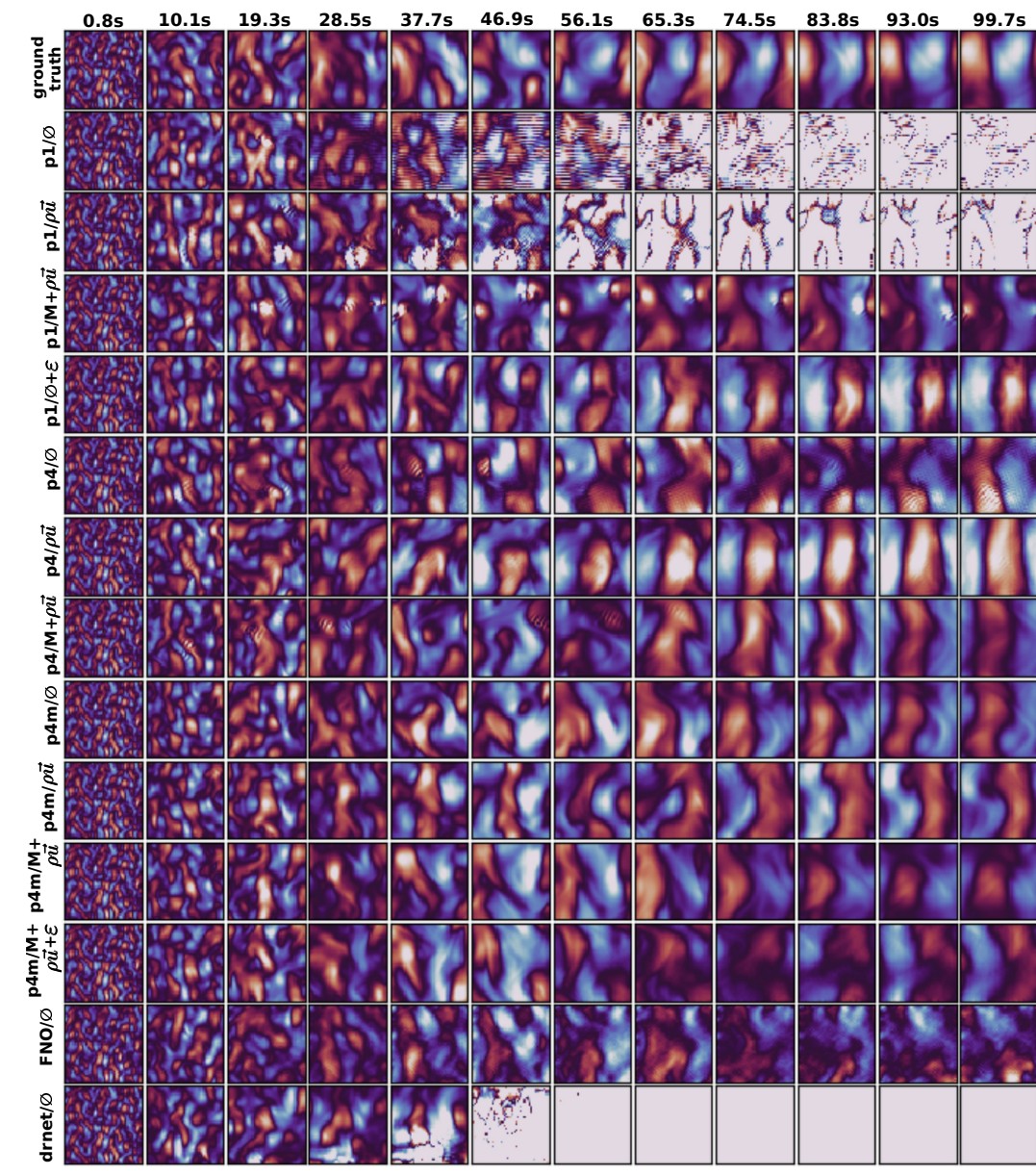

Figure 17: An example of rollout predictions on $v$ from all methods with network sizes of approximately 0.1M parameters for the incompressible Navier-Stokes equations at different time steps. The top row shows the ground truth as a reference.

Figure(18) presents the velocity and energy spectra over additional time steps, expanding on the two time steps shown in Figure (4-d,e,f,g) of the main text. Our analysis shows that the best-performing model, p4m/M+$\rho\vec{u}$ (blue curve), consistently provides the closest match to the reference across all time steps for both the velocity and energy spectra.

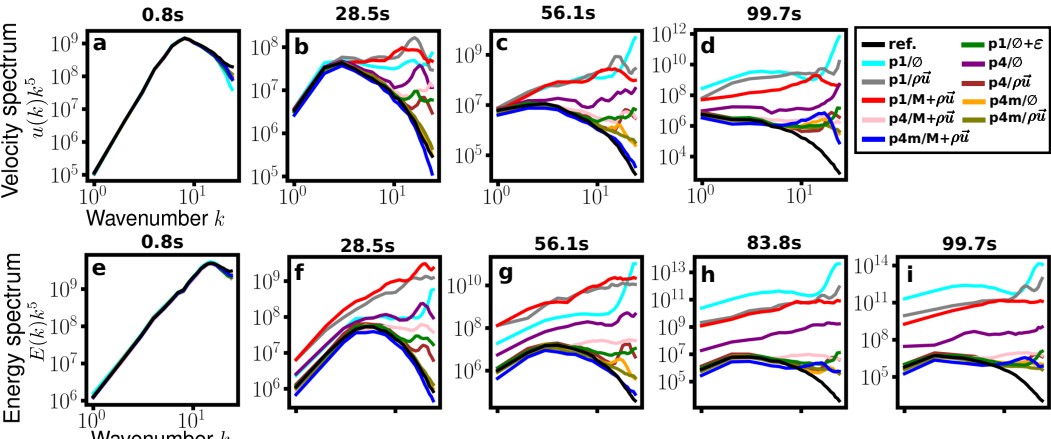

Figure 18: The velocity and energy spectra at additional time points correspond to Figure (4-d,e,f,g) in the main text. Our best method, p4m/M+$\rho\vec{u}$, closely matches the reference spectra, shown as black curves.

We also report the NRMSE and correlation over long rollouts on the field variable $u$ for the decaying turbulence case. The models used are p4m/M+$\rho\vec{u}$ (our best performing model) and p1/$\varnothing$, both trained with clean data and training noise $\mathcal{N}(\mu = 0, \sigma = 0.001)$. We find that the model p4m/M+$\rho\vec{u} + \varepsilon$ which is trained with noise, reaches a lower accuracy but higher rollout correlation compared to its counterpart trained with no noise p4m/M+$\rho\vec{u}$.

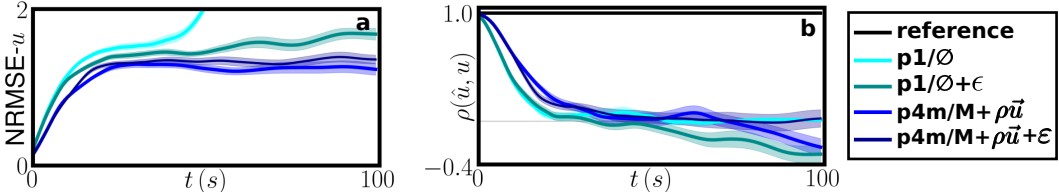

Figure 19: The comparison of models trained with noise and clean data: p1/$\varnothing + \varepsilon$, p4m/M+$\rho\vec{u} + \varepsilon$, p1/$\varnothing$ and p4m/M+$\rho\vec{u}$. We find that training with noise provides longer rollout stability but lower accuracy compared to training with clean data.

## H.3 THE DETAILS ON GENERALIZATION TASKS

The test of generalization for shallow water equations, we focus on two experiments: (1) one rectangle $\zeta = 0.1m$ as IC with random size and location; (2) two rectangles $\zeta = 0.1$ with random size and location. Thus, in later experiment, one rectangle can cover to another one to generate a new shape, for example, a "L" shape illustrated in main test. In the cover case, the cover domain $\zeta = 0.2$ which is the sum of two rectangles. Therefore, it is also a more challenging and general case.

Figure 20 presents the generalization results of the SWE for all models, using the L-shaped surface elevation IC. Our best model is p4m/M with p4m/∅ a close second.

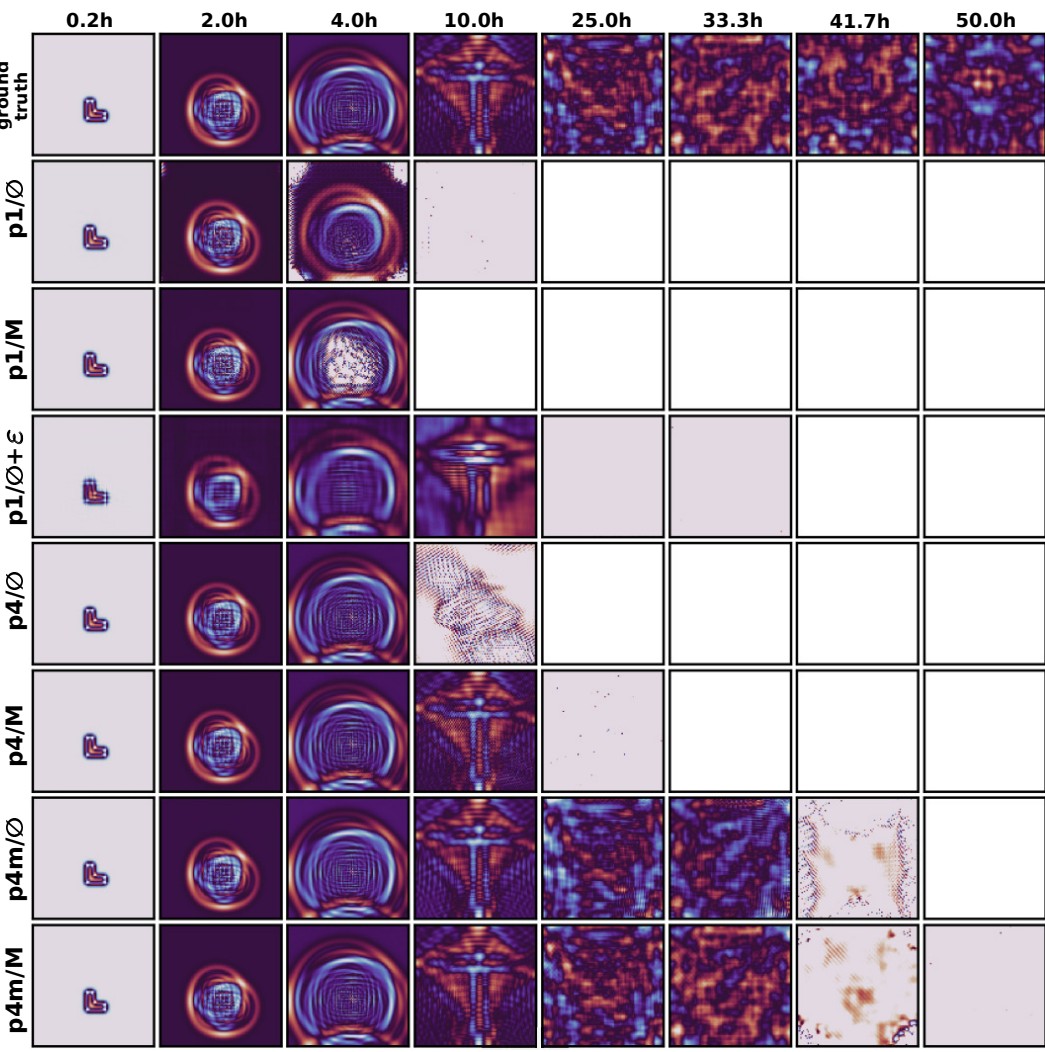

Figure 20: The rollouts demonstrating generalization for the SWE, obtained from all methods with an L-shaped IC, are shown at various time intervals. This is the detailed version of the (5-a) from the main text

Fig. 21 presents the generalization results of the SWE for all models, using a rectangular-shaped elevation IC. In this case, p4m/M, outperforms all other methods in both accuracy and long-rollout prediction compared to the ground truth. Our best model, p4m/M, accurately predicts the surface elevation $\zeta$ up to 50 hours, while all other models fall short, with predictions failing before 25 hours.

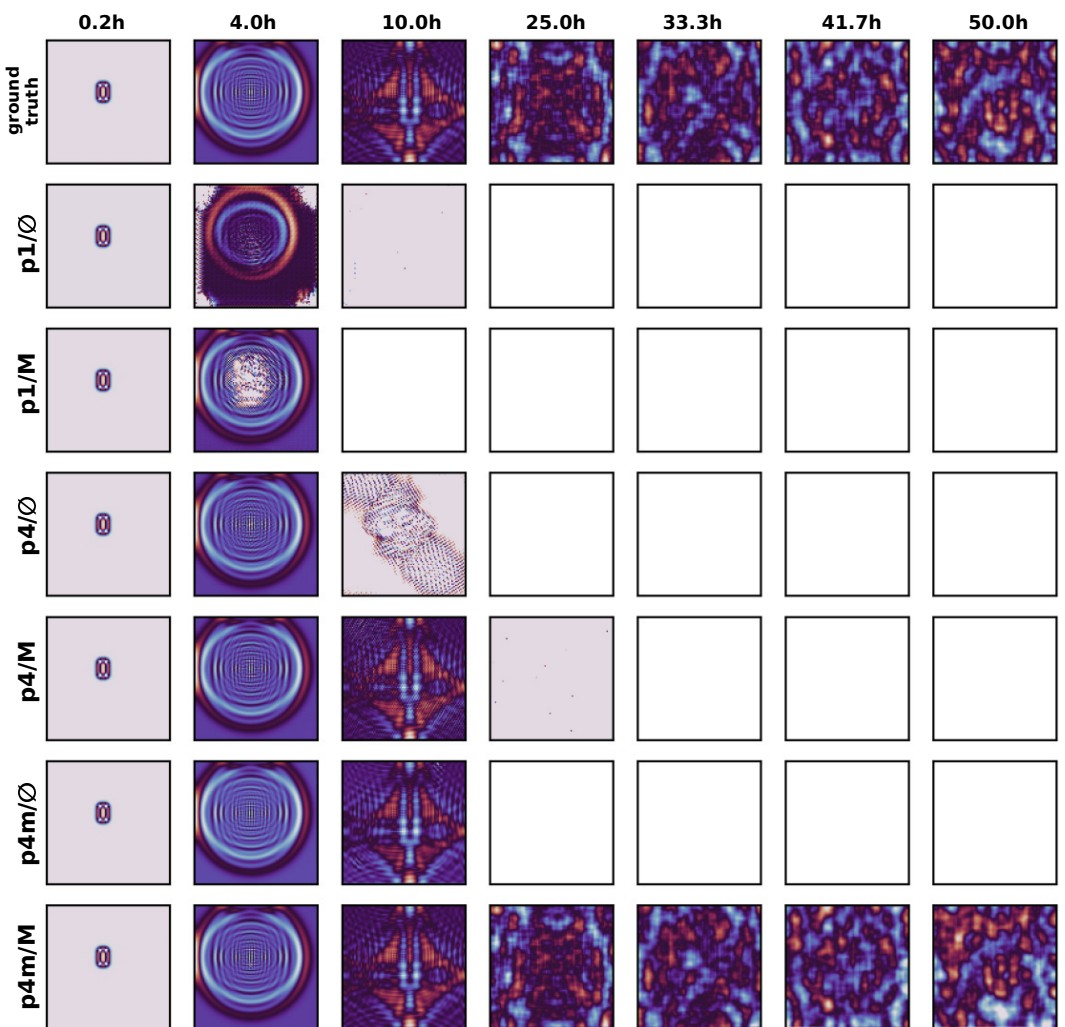

Figure 21: The rollouts demonstrating generalization for the SWE, generated by all methods using a single rectangular-shaped elevation IC, are shown at various time intervals.

Fig. 22 illustrates the generalization of rollout performance for the SWE with ICs of two rectangular-shaped elevations. This is particularly a challenging problem because we train using experiments with ICs of single square-shaped elevation. We find that p4m/M achieves the best rollout performance and correctly predicts the surface elevation for time 25h rollouts.

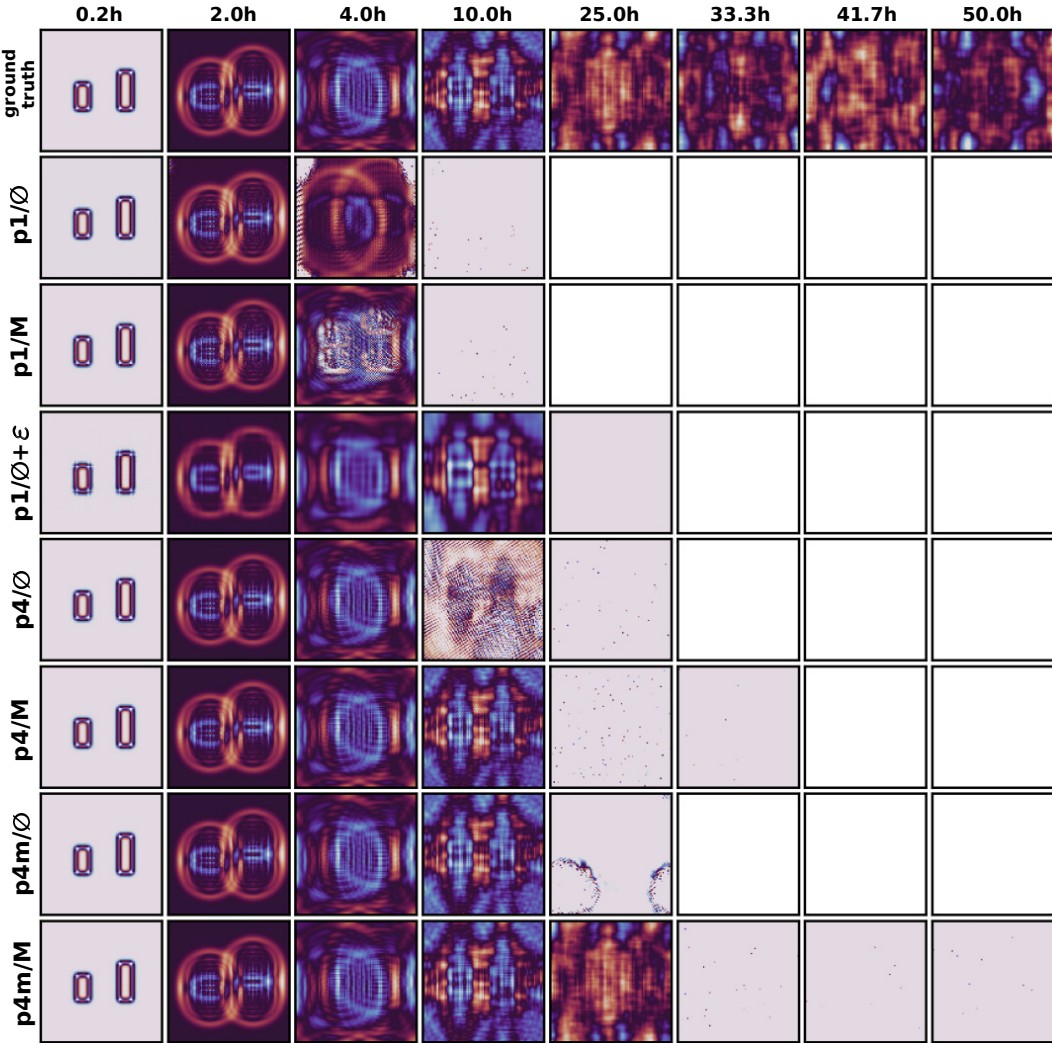

Figure 22: The rollouts demonstrating generalization for the Shallow Water Equations (SWEs) on rollout performance, from all methods are shown at various times, using a challenging IC: two rectangular-shaped elevations. p4m/M achieves the best rollout performance.

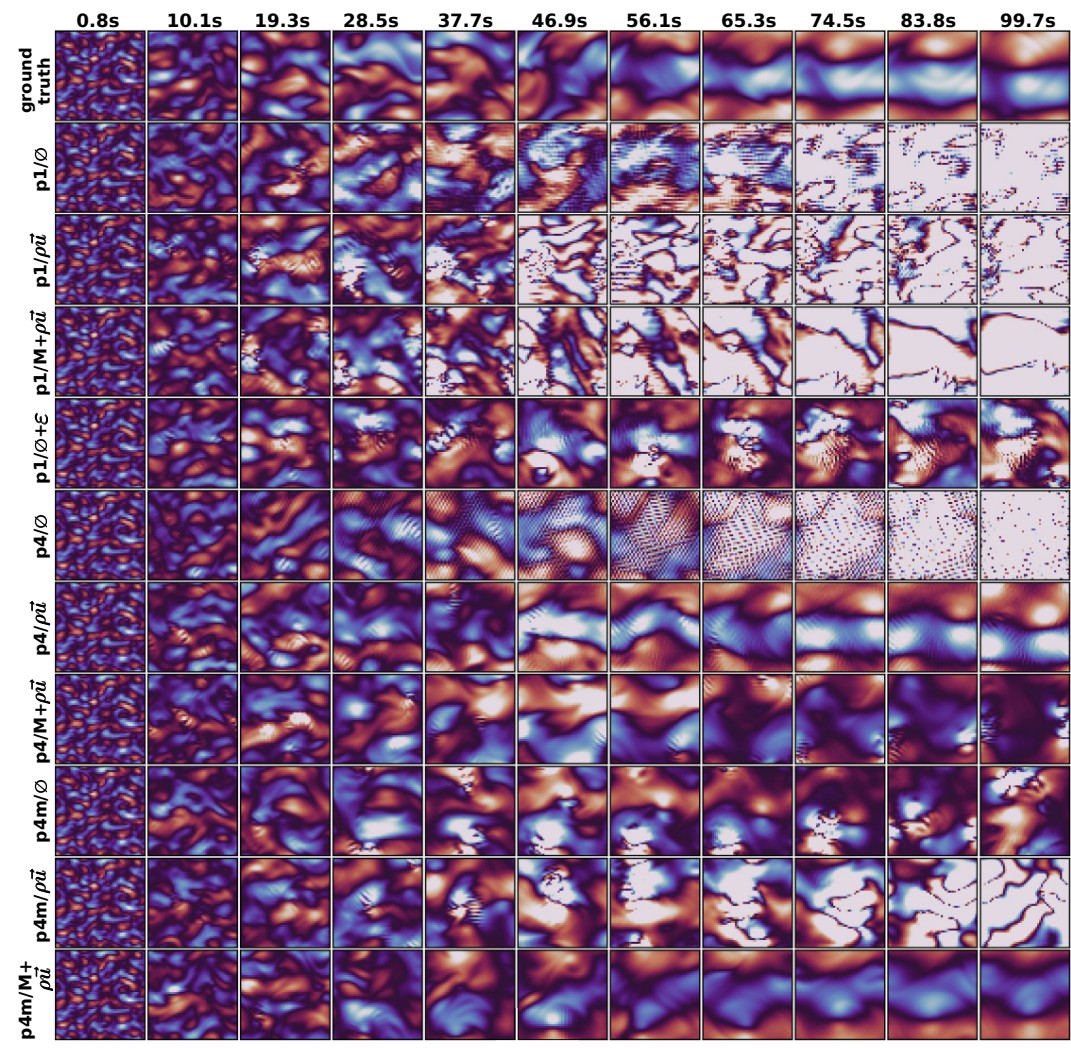

Figure 23: The rollout performance of networks with different physical and symmetry constraints for the decaying turbulence case. All plots show the evolution of the field variable $u$. This is the detailed version of the Figure (5-d) from the main text. It indicates that the model p4m/M+$\rho\vec{t}$ aligns more closely with the ground truth trajectory and remains stable over a longer period compared to other models

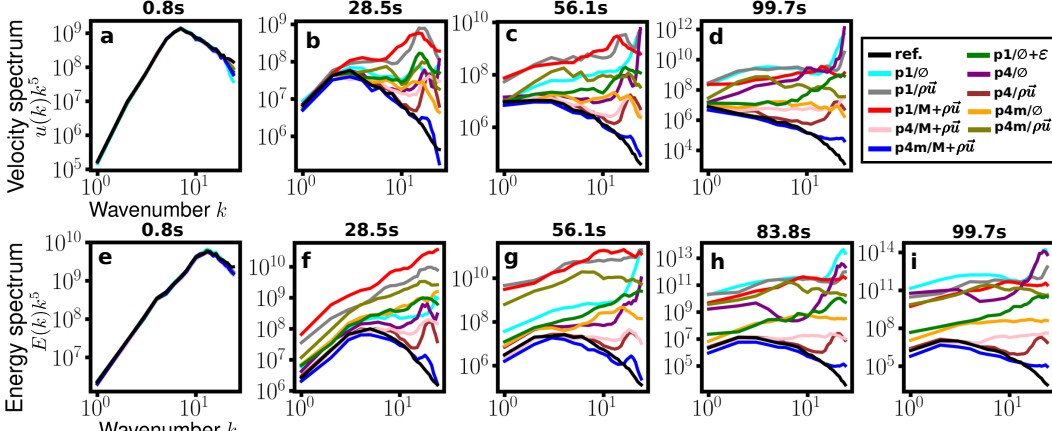

Figure 24: The detailed analysis of velocity $u$ and energy power spectra over longer rollouts for the decaying turbulence case is presented. This expands the Figure 5-(e,f) from the main text. We find that the best-performing network is p4m/M+$\rho\vec{u}$, it matches the energy and velocity spectra closer compared to networks with other constraints.

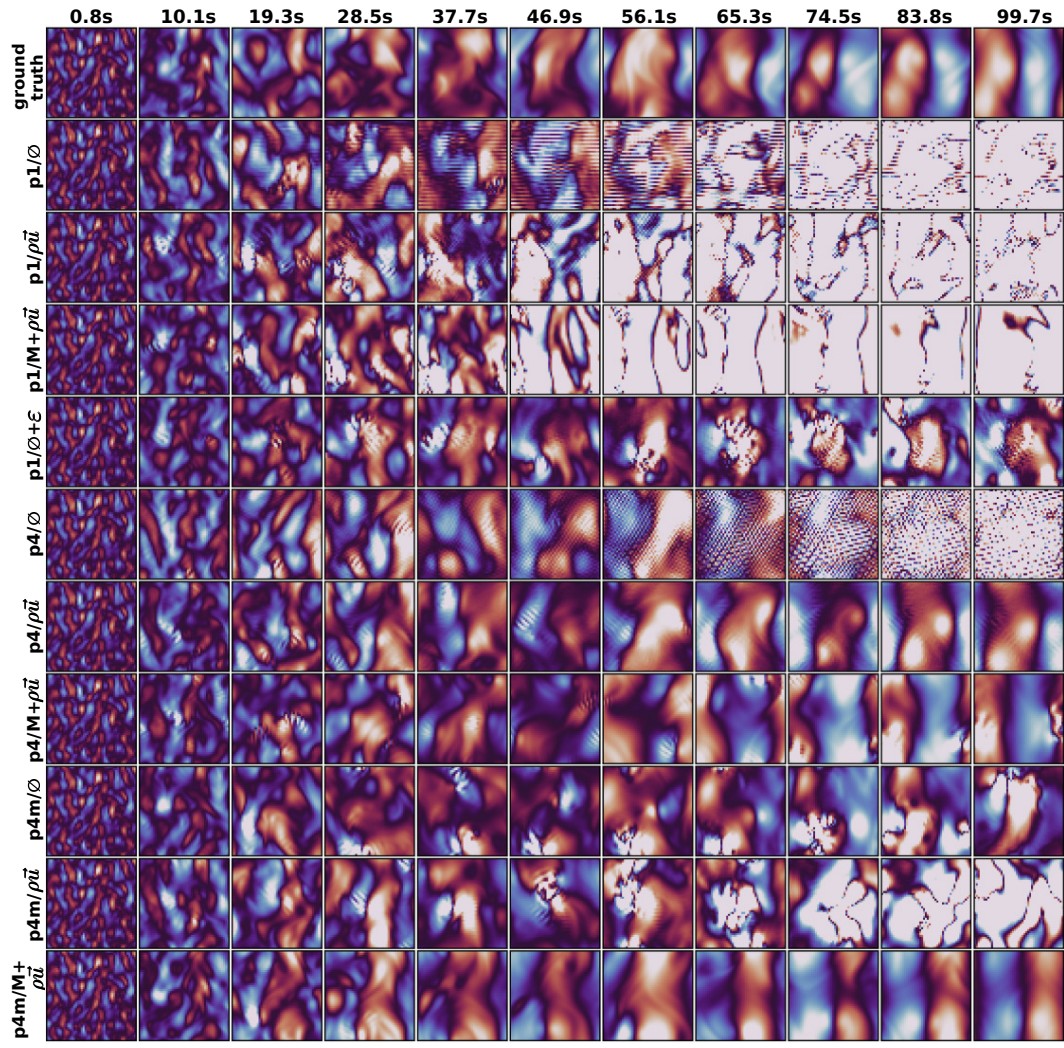

Figure 25: The rollout performance of networks with different physical and symmetry constraints for the decaying turbulence case. All plots show the evolution of the field variable $v$. This is the detailed version of the Figure (5-d) from the main text.

## H.4 DETAILS ON THE EFFECTS OF NETWORK AND DATASET SIZE ON ROLLOUT PERFORMANCE

Fig. 26 depicts a detailed analysis of the effect of network size and training data size on rollout performance. We examine three different network sizes: 0.1M, 2M, and 8.5M parameters, and three different training dataset sizes: 100, 400, and 760 experiments with varying ICs. Increasing the network size or the training dataset size improves the rollout performance of the network. Additionally, the network with physical and symmetry constraints performs better in each case.

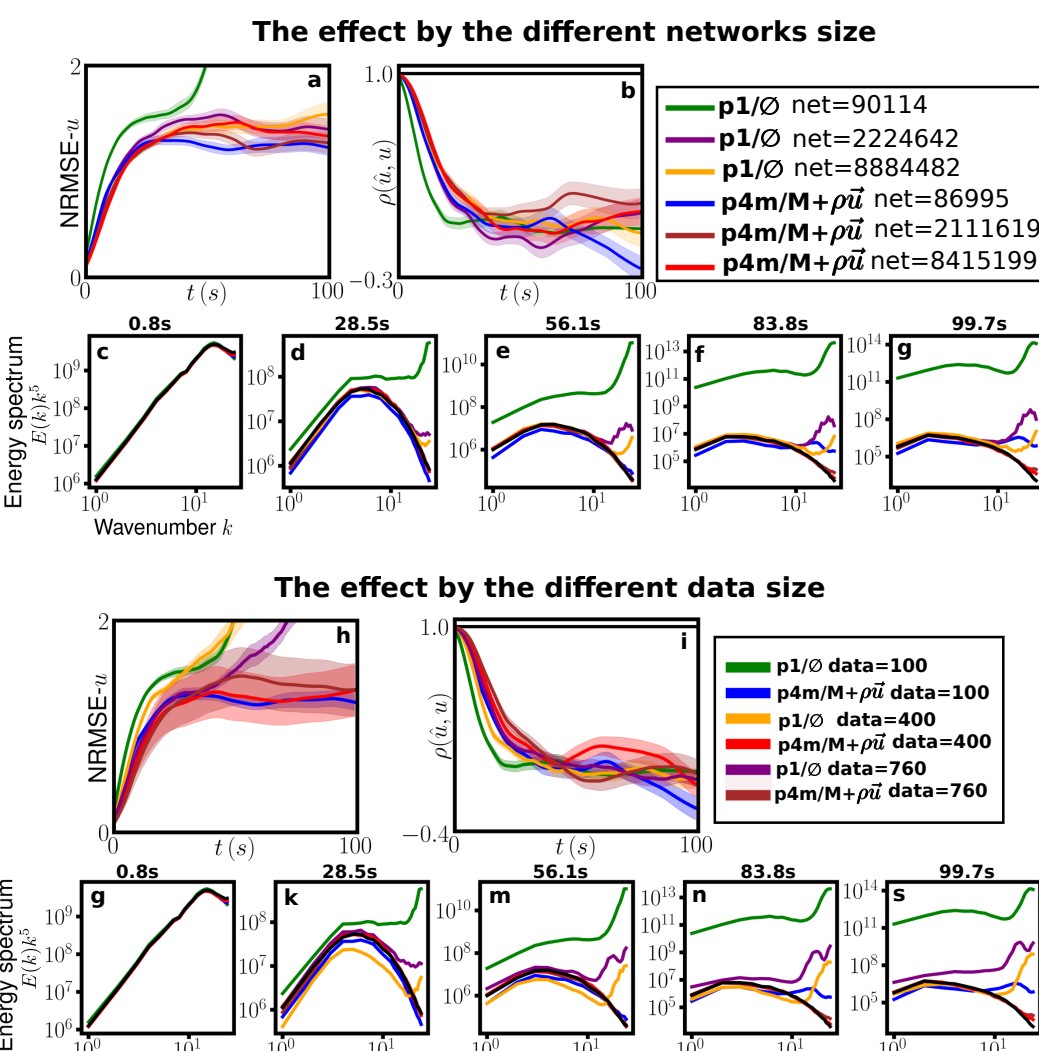

Figure 26: Top: Influence of network size on NRMSE-$u$, $\rho(\hat{u}, u)$ (a, b) and energy spectrum (c, g) for p1/$\varnothing$ and p4m/M+$\rho\vec{u}$. Bottom: Influence of training data size on NRMSE-$u$, $\rho(\hat{u}, u)$ (h, i), and the energy spectrum (g,s). All results are reported on $99.7s$ rollouts.

