# OpenReview forum: "Geometric and Physical Constraints Synergistically Improve Neural PDE Integration"
_ICLR.cc/2025/Conference — Submitted to ICLR 2025_

### Official Review · Reviewer_qgvH · 2024-10-28

**Soundness:** 3
**Presentation:** 3
**Contribution:** 3
**Rating:** 6
**Confidence:** 5

**Summary:**

I thank the authors for an interesting submission. I found the central idea clear.

It concerns itself with building in the inductive bias of symmetry constraints and conserved quantity preservation directly into the structure of neural PDE surrogates. They explore the p4m group of discrete translations, reflections, and right-angle rotations. Most importantly, they build these symmetries into a model that maps between staggers Arakawa C-Grids, the dominant modality for modeling fluids on regular domains. The experiments are an ablation of this idea on decaying turbulence and shallow water equations, which are common datasets/systems of consideration in this field. As expected, the authors find that increasing amounts of constraint, lead to better models. This is because in this field the constraints on preserved quantities and symmetries are exact.

**Strengths:**

Novelty and significance
- As far as I am aware there this is novel work. I do not know of examples of equivariant layers in the neural PDE literature that concern themselves with staggered grids as used for CFD. The Arakawa C-grid is the most commonly used in CFD applications and so this work is also aiming for impact in areas of practical significance.

Written quality
- The written quality of the submission is high. The style is clear, the objective is clear, the exposé of ideas and execution is clear.

Experiments
- The experiments on the shallow water equations (Fig 3) show that adding the symmetry constraints improves decorrelation time compared to baselines without.
The experiments on decaying turbulence (Fig 4) show that the adding symmetry constraints improves energy spectrum in the far term (chaotic field)
- I like the ablation where the authors try an out-of-distribution test in Figure 4. It highlights the importance of inductive biases for out-of-distribution reasoning

**Weaknesses:**

Significance and scope
- The work is restricted to considering finite symmetry groups of discrete grids (the p4m group of translations, reflections, and $90^\circ$ rotations, and subgroups thereof). While the symmetry groups under consideration are exact (and probably a lot simpler to implement than groups of continuous actions), they are indeed rather limited in scope.
- It would have been nice to see explorations in other groups. PDEs have quite interesting Lie point symmetry groups e.g. Brandstetter et al. (2022) and Wang et al. (2021) and it would have been nice to show that equivariance to these groups can be achieves on the grid.

Experiments
- The greatest weakness of an otherwise solid paper is a lack of comparisons with contemporary works in the field that either:
- a) explore the imposition of symmetries on to neural PDE surrogates for instance _“Incorporating Symmetry into Deep Dynamics Models for Improved Generalization”_ (Wang et al., 2021) or in the very least use data augmentations as in _“Lie Point Symmetry Data Augmentation for Neural PDE Solvers”_ (Brandstetter et al., 2022) or
- b) explore other kinds of stabilization techniques, such as noise injection _“Learned Coarse Models for Efficient Turbulence Simulation”_ (Stachenfeld et al., 2021), pushforward technique _“Message Passing Neural PDE Solvers”_ (Brandstetter et al., 2022), alternative stabilizing losses such as _DySLIM_ (Schiff et al., 2024) or denoising architectures such as _PDE-Refiner_ (Lippe et al., 2023).

I do think the paper is well-motivated, written well, and within the (limited) experimental scope convincingly performant, but the lack of experimental baselines from elsewhere in the field is of concern when it comes to rigor. I believe cannot encourage the authors to run more experiments at this point, but I do want to highlight that this is the main reason why I can only support this paper as a marginal accept at most.

**Questions:**

- Equation 73: You enforce mass conservation by removing a global mean. This is somewhat different to the way that FVM works, where one first computes fluxes through cell faces and then applies continuity within a cell to compute the change of mass in each cell. 1) Did you try the former FVM-like update rule and 2) would the FVM-like rule not exemplify the use of the staggered grid more, after all the output layer computes quantities of the faces of the grid?
- How does this work tie into classical methods such as equivariant moving frame theory in _“Symmetry Preserving Numerical Schemes for Partial Differential Equations and their Numerical Tests”_ by Rebelo & Valiquette (2011). That paper shows how to construct finite difference schemes based of the Lie Points Symmetry group.
- Nit: some of the references are not placed in brackets; for example, line 93 or 105 or 115. I would advise the authors to check for these (I think the missing LaTeX command is \citep{})
- Nit: Tables 1 and 2. I found it confusing that m denotes reflection in p4m and also mass. Maybe try a different symbol?
Nit: line 347: perdition -> prediction

---

> ### Author Response · Authors · 2024-11-28
>
> > Significance and scope
> >
> > The work is restricted to considering finite symmetry groups of discrete grids (the p4m group of translations, reflections, and rotations, and subgroups thereof). While the symmetry groups under consideration are exact (and probably a lot simpler to implement than groups of continuous actions), they are indeed rather limited in scope. It would have been nice to see explorations in other groups. PDEs have quite interesting Lie point symmetry groups e.g. Brandstetter et al. (2022) and Wang et al. (2021) and it would have been nice to show that equivariance to these groups can be achieves on the grid.
>
> See response to reviewer 2's final point above. We agree that further exploration would be desirable, though we find our paper is already quite "full" with detailed results and analyses of our two tasks, many constraints and scaling properties. We also chose to focus on exact equivariance, taking into account the grid and discretizaiton scheme, to remove ambiguity as to whether symmetry constraints were being fairly evaluated.
>
> > The greatest weakness of an otherwise solid paper is a lack of comparisons with contemporary works in the field that either:
> >
> > a) explore the imposition of symmetries on to neural PDE surrogates for instance “Incorporating Symmetry into Deep Dynamics Models for Improved Generalization” (Wang et al., 2021) or in the very least use data augmentations as in “Lie Point Symmetry Data Augmentation for Neural PDE Solvers” (Brandstetter et al., 2022) or
> >
> > b) explore other kinds of stabilization techniques, such as noise injection “Learned Coarse Models for Efficient Turbulence Simulation” (Stachenfeld et al., 2021), pushforward technique “Message Passing Neural PDE Solvers” (Brandstetter et al., 2022), alternative stabilizing losses such as DySLIM (Schiff et al., 2024) or denoising architectures such as PDE-Refiner (Lippe et al., 2023).
>
> Regarding point (a), we wish to point out that the architecture implemented in Wang et al. 2021 cannot be fairly applied to our tasks, since it cannot handle staggered grids without breaking equivariance. This was a direct motivation for our study. Furthermore, Wang et al. essentially implemented equivariant "vanilla" U-nets and Resnets, which exhibited similar performance to each other, and could not be fairly compared to our more recent "modern" U-nets (see response to reviewer 2 above). We emphasize that our aim was not to produce state of the art results on these tasks by any means, but rather to evaluate the utility of physical and geometric constraints, alone and together. We also now cite the suggested paper on data augmentation in our discussion.
>
> Regarding point (b), we did include a comparison to noise injection in our paper, which now appears in fig. 3-5, as well as many figures in the appendix. Models trained with noise injection are marked with a $\varepsilon$ in figure legends. Noise injection improved results but was not as effective as physical or geometric constraints. We now mention pushforward, PDE-refiner and DySlim as alternative approaches in the discussion.
>
> >equation 73: You enforce mass conservation by removing a global mean. This is somewhat different to the way that FVM works, where one first computes fluxes through cell faces and then applies continuity within a cell to compute the change of mass in each cell. 1) Did you try the former FVM-like update rule and 2) would the FVM-like rule not exemplify the use of the staggered grid more, after all the output layer computes quantities of the faces of the grid?
> How does this work tie into classical methods such as equivariant moving frame theory in “Symmetry Preserving Numerical Schemes for Partial Differential Equations and their Numerical Tests” by Rebelo & Valiquette (2011). That paper shows how to construct finite difference schemes based of the Lie Points Symmetry group.
>
> We agree that a flux-conservative update would be more similar to FVMs, and as a fully convolutional method would allow changing the domain size after training. We believe this is possible but have so far not succeeded in building such an output layer, we have added this point to our discussion. We appreciate the reference to Rebelo & Valiquette, of which we were unaware and have now added to the manuscript.
>
> > Nit: some of the references are not placed in brackets; for example, line 93 or 105 or 115. I would advise the authors to check for these (I think the missing LaTeX command is \citep{})
>
> Fixed as suggested.
>
> > Nit: Tables 1 and 2. I found it confusing that m denotes reflection in p4m and also mass. Maybe try a different symbol?
>
> Changed as suggested.
>
> > Nit: line 347: perdition -> prediction
>
> Fixed as suggested.

---

> > ### Comment · Reviewer_qgvH · 2024-11-29
> >
> > Thanks for the updated references and effected suggested changes.
> >
> > I have a query to make about the differences between UNets and modern UNets. Do you have a reference of your own or in the ML4PDEs literature providing a quantitative comparison on some standard benchmarks between the two? This is just to get an idea of the gap in performance. I ask this since you claim that these models cannot be compared, but as far as I can tell the basic model is still the same apart from a few extra skip connections and some minor architectural details.
> >
> > I would also like to ask exactly how is it that a UNet would break equivariance if applied to a staggered grid?

---

> > > ### Author Response · Authors · 2024-11-29
> > >
> > > The comparison between the original U-net and a modern U-net optimized on PDE tasks can be found in [Gupta & Brandstaetter, 2022](https://arxiv.org/abs/2209.15616). The tables in the supplementary material given the fullest comparison on SWE and Navier Stokes tasks (not the same as our tasks, as these versions did not use mass- or momentum-conserving solvers and used differents BCs). Code implementations for U-net variants accompany this paper, and an independent implementation for modern U-net with some minor variations (e.g. ConvNext blocks) can be found in [Kohl et al., 2023](https://arxiv.org/abs/2309.01745).
> > >
> > > There is nothing specific to the original or modern Unets that would break equivariance on a staggered grid -- rather, any existing input layer that does not take the grid into account will break equivariance. We therefore could not fairly compare any existing U-net implementation to our results on our task, since these lack support for C-grids and would not be equivariant. We chose to modify the modern U-net, which has become a standard strong baseline for subsequent improvement as a neural PDE surrogate, for example through generative modeling (Kohl et al. 2023) or PDErefiner ([Lippe et al. 2023](https://arxiv.org/abs/2308.05732)). We thus combined our novel input/output layers with the existing modern U-net architecture. We could have done the same for the original U-net, ResNet or other architecture. If we were to devote additional time and resources toward creating an equivariant version of a second existing architecture, we would likely not choose the 2015 U-net but rather a Dilated ResNet, or possibly an attention-based architecture. We note that our manuscript now compares to FNO as a baseline, though we do not have an equivariant version of this architecture.

---

> > > > ### Comment · Reviewer_qgvH · 2024-12-02
> > > >
> > > > Thanks for the additional information.
> > > >
> > > > It appears the UNet and modern UNet are only very slightly different in performance, but now there is precedent for the modern UNet in this field.
> > > >
> > > > And thanks for explaining how not accounting for the C-Grid would break equivariance

---

### Official Review · Reviewer_8dXp · 2024-10-30

**Soundness:** 2
**Presentation:** 2
**Contribution:** 2
**Rating:** 3
**Confidence:** 4

**Summary:**

The authors present a method for enfiorcing symmetry and PDE constraints in neural PDE "integration", together with a study on how symmetry constraints and physical constraints act together to improve the rollout stability for the investigated PDE settings.

While I find this setting thematically intriguing and while I do see merit in the proposed method, I find the manuscript lacking on multiple fronts. My main issue is misleading messaging, as the title suggests a much broader work and a systematic treatment that would go beyond 2 examples and a single architecture. Moreover, the explanation of the method is quite lacking in my view and I would have wished for either more figures and/or a better explanation of the method in the text.

**Conclusion**
While I think that this is promising work, I wish that the treatment was better explained and more systematic. Enforcing constraints can be done in many ways and this is just a single one with a single baseline. The title suggests a much broader study and I think the literature would benefit from a systematic study that studies the importance of physical and symmetry constraints.

As such, I do find the manuscript to be incomplete. In a normal Journal article I would suggest a major revision, but as this is a Conference venue, I suggest tthe authors rework the article and resubmit to the next conference.

**Strengths:**

The authors suggest an interesting method for constraining neural PDE surrogate models and show how it improves the rollout stability for their baselines. They present 2 non--trivial examples on a staggered grid, which is a departure from standard regular grids in PDE surrogates.

**Weaknesses:**

**Major**

- **Misleading messaging**: I have an issue with the term "PDE integration" and as a consequence, the title, abstract and large parts of the introductory text are misleading. Unless the PDE operator is integrated numerically, by discretizing it, we cannot really call this PDE integration. We are rather learning neural surrogates, which in some regime may approximate the integral solution operator. But we are not integrating, in contrast to neural ODEs for examples, which I initially thought were related. I find the title of the paper is highly misleading in this regard and it over-sells what is actually found in the paper. I would strongly recommend the authors revise title, abstract and introduction and adapt their messaging
- **Lacking explanation of the proposed method**: The method is not explained in detail and from the text alone and Figure 1 it is hard to understand how symmetries and/or physical constraints are exactly enforced. While I do see that the text refers to Appendix C, the formulas in the appendix do not help to convey the core idea of these layers. As such, I would propoise the authors to rework the explanation of the method in the main text and augment it with figures
- **Weak baselines**: The method is only compared to a "vanilla" U-Net, which is not very well suited to autoregressivey approximate solution operators of PDEs. It would be great to have other baselines e.g. FNO and it's physics-constrained counterpart: PINO which also enforces physicality during training.

**Minor**

- I find Figures 2,3,4 quite overloaded and hard to read. Distributing these figures into separate Figures would make the paper much easier to parse and also drastically reduce the amounto f explanation these Figures require
- Why is Section 4.2 called Decaying turbulence? I do not see any mention of turbulence in this section. Is this simply due to the chosen example? Is the fact that it is decaying somehow important

**Questions:**

I would be curious how exactly physical constraints are enforced in the output layer. From the text and figures alone, this is unclear to me. Moreover I am curious why not more PDEs were tested such as non-hyperbolic ones? Wouldn't this method be applicable there as well? Also what about PDEs on manifolds with more complex symmetries, such as SO(3)-equivariance?

---

> ### Author Response · Authors · 2024-11-28
>
> >Misleading messaging...
>
> We must agree with the reviewer on this point. We have thoroughly revised the relevant language including the title, abstract and introduction as suggested. While the literature on ML-based prediction of PDE system states uses inconsistent nomenclature, with various papers referring to integrating, solving, predicting or modeling the PDE, we now refer to neural surrogates that predict future PDE states.
>
> >Lacking explanation of the proposed method:...
>
> We have revised the main text, figure captions and supplementary materials to provide additional detail. We have added an additional figure (fig. 2) showing the input layers in greater details. We could not introduce further figures due to space constraints, but refer to existing figures in the additional text for explanation purposes.
>
> >Weak baselines: The method is only compared to a "vanilla" U-Net, which is not very well suited to autoregressivey approximate solution operators of PDEs. It would be great to have other baselines e.g. FNO and it's physics-constrained counterpart: PINO which also enforces physicality during training.
>
> We agree that a "vanilla" U-net (Ronneberget et al., 2016) is not ideal as a PDE surrogate, as this architecture was originally introduced for segmentation of medical images. However, we have not used this version, but rather a "Modern U-net" (Gupta et al. 2022, Kohl et al. 2024), specifically optimized for use as a PDE surrogate. The difference in performance as a PDE surrogate between the original and modern U-nets is evaluated in Gupta et al., 2022. We now clarify this distinction in section 3.1. We have also added an FNO baseline.
>
> >I find Figures 2,3,4 quite overloaded and hard to read. Distributing these figures into separate Figures would make the paper much easier to parse and also drastically reduce the amounto f explanation these Figures require
>
> We agree that understanding these figures was somewhat difficult. While space constraints prevented us from splitting them into additional figures, we have condensed the relevant captions and references to the figures in the main text, and carefully revised to increase clarity.
>
> > Why is Section 4.2 called Decaying turbulence? I do not see any mention of turbulence in this section. Is this simply due to the chosen example? Is the fact that it is decaying somehow important
>
> The terminology "decaying turbulence" was introduced together with this task in Kochkov et al., 2021. Turbulent eddies are resolved by the DNS simulation used for data generating. The fact that these decay over time to form larger, smoother structures makes the task more challenging as the distribution of velocity fields is not constant over time. We now clarify these details in section 4.2.
>
> > I would be curious how exactly physical constraints are enforced in the output layer. From the text and figures alone, this is unclear to me.
>
> Mass conservation for SWE and momentum conservation for INS were inforced by subtraction of the global mean from learned update fields, before applying these updates to the velocity field ($u, v$, INS) or surface height perturbation field ($\zeta$, SWE). To enforce both mass and momentum conservation for INS, we used the Helmholtz decomposition to write updates $\Delta u, \Delta v$ as the curl of a periodic streamfunction $\nabla \times a$. We now discuss these strategies in greater detail in section 3 and appendix C.
>
> > Moreover I am curious why not more PDEs were tested such as non-hyperbolic ones? Wouldn't this method be applicable there as well?
>
> We agree that testing additional PDEs would be worthwhile, and can see no reason why our techniques would not apply to hyperbolic PDEs. Given the highly detailed experiments (multiple levels of geometric and physical constraints, various network size and dataset) we find this to be beyond the scope of the current work. However, we now acknowledge this limitaiton and discuss the possible extension to hyperbolic PDEs in the discussion.
>
> > Also what about PDEs on manifolds with more complex symmetries, such as SO(3)-equivariance?
>
> The use of discretized grids means that the time stepping operator $\mathcal G$ is not equivariant to continuous symmetry groups, but only to discrete symmetries of the grid, which for 2D square grids are limited to reflections, 90 degree rotations and translations (p4m and its subgroups). Nevertheless, the streerable CNN layers we build on are approximately covariant to continuous reflections (Cesa et al., 2022), and we have now included this topic in our discussion, along several new references.

---

### Official Review · Reviewer_wqWf · 2024-11-01

**Soundness:** 3
**Presentation:** 3
**Contribution:** 2
**Rating:** 3
**Confidence:** 4

**Summary:**

This paper systematically investigates how various symmetries and conservation laws affect PDE integration accuracy, individually and in combination. To impose these as hard constraints, this paper introduces novel input and output layers. Exepriments are carried on two challenging tasks: shallow water equations with closed boundaries and decaying incompressible turbulence, and it was shown that both types of constraints improve performance consistently across integration times, accuracy measures, and network sizes, and that symmetries are more effective than physical constraints and optimal performance was achieved by combining both.

**Strengths:**

1. Extensive experiments on two challenging tasks and state-of-the-art performance on prediction accuracy.
2. clear conclusion that symmetries are more effective than physical constraints and optimal performance was achieved by combining both.
3. Introduction of novel input and output layers to staggered grids to impose symmetry and conservation constraints.

**Weaknesses:**

1. Novelty.  This paper focuses on testing the effect of symmetries and conservation laws on PDE integration accuracy to give a transparent comparison of the effectiveness of individual techniques. Enhancing neural PDE integration with symmetries has been explored in (Wang et al., 2020; Helwig et al., 2023; Smets et al., 2023; Huang & Greenberg, 2023; Ruhe et al., 2024), and improving neural PDE integration with physics constraints has been explored through additional loss terms (Read et al., 2019; Wang et al., 2020; Stachenfeld et al., 2021; Sorourifar et al., 2023), or by reparameterizing neural network outputs to respect them as hard constraints (Mohan et al., 2020; Beucler et al., 2021; Chalapathi et al., 2024; Cranmer et al., 2020; Greydanus et al., 2019). Besides novel input and output layers and a fair comparison, are there any algorithmic novelties in comparison to these works?
2. In fig.3 and fig.4, the energy is shown. Why did not show the mass directly which is the conserved quantity under consideration?
3. Generalization: the SWE equation is linear to the fluid surface elevation , can this fact explain the generalization capability on SWEs using a "L" shaped initial surface? How to explain the generalization capability on INS? Also, please give a definition or reference for wavenumber, which did not appear in equation (9).
4. In the paragraph starting from line 187, please cite appendix E that describes the phyiscs constraints. And in equation (73), what is the relationship between mass and the surface elevation?
5. Please give the hardware used and simulation time to generate the reference solution and training data. Also, please give the hardware used and GPU-hours for training.
6. Typos: line 68, "to solutions at time via supervised learning" should be "to solutions at time t via supervised learning"? line 256, "Iinitial" should be "Initial", line 347, "perdition" should be "prediction".
7. Theory. This paper could be improved from the theoretical aspect. For example, the conserved physical quantities are related quantitatively with the solutions, so the losses corresponding to solution prediction and conservation laws are correlated, and maybe a loss bound or convergence rate can be analyzed based on this.

**Questions:**

see weaknesses.

---

> ### Author Response · Authors · 2024-11-28
>
> > Novelty...
>
> We acknowledge that our manuscript does not introduce new algorithms or complete new architectures, but argue that it nonetheless contains important empirical findings. These include:
> * The effectiveness of combining physical and geometric constraints.
> * The comparison of these techniques with alternatives (such as FNOS or training with input noise).
> * The evaluation of these benefits across dataset and architecture sizes, and across tasks.
>
> Fundamentally, machine learning researchers need better tools to control the long-term behavior of trained dynamical models, and we would ask the reviewers to consider the novelty and relevance of results validating a strategy for extending the duration of accurate rollouts. This is particularly relevant for the rapidly developing field of neural PDE surrogates, and its applications to real-world applications such as weather and climate-related predictions.
>
> > In fig.3 and fig.4, the energy is shown. Why did not show the mass directly which is the conserved quantity under consideration?
>
> We agree conservation laws should be directly evaluated, and now include this mass in the SWE figure (now fig. 3), and all other quantities in the supplememntary materials.
>
> > Generalization: the SWE equation is linear to the fluid surface elevation, can this fact explain the generalization capability on SWEs using a "L" shaped initial surface? How to explain the generalization capability on INS?
>
> The SWEs in eq. 7-8 are linear in fluid surface elevation $\zeta$ when the velocity field $u$ is fixed, but the both the implicit time stepping and the presence of nonlinear PDE terms (the product $hu$ and bottom drag) prevent linear interpolation of the time-integrated PDE solutions.
>
> The generalization and interpolation capabilities of neural networks is an active area of research beyond the scope of this work, but we can speculate that the constraints imposed on the network -- translation equivariance for convolutions, other symmetries, and physical constraints, can improve generalization performance by ruling out extensions of the learned time stepping function beyond the training data that would violate our physical understanding or the PDE symmetries. We thank the reviewer for pointing out the relevance of this topic have added additional material on this point to our discussion section.
>
> > Also, please give a definition or reference for wavenumber, which did not appear in equation (9).
>
> Wavenumbers appear in two contexts: the peak wavenumber used to define initial conditions for INS, and as the x-axis for velocity and energy spectra. We now give clear definitions of both usages.
>
> > In the paragraph starting from line 187, please cite appendix E that describes the physics constraints.
>
> We now cite appendix E, describing how physical constraints are imposed, in this paragraph.
>
> > And in equation (73), what is the relationship between mass and the surface elevation?
>
> Mass in a grid column is directly proportional to the total surface height $m=\rho h (\Delta x)^2$ where $\rho$ is density and $h = \zeta + d$. Thus conservation of mass is equivalent to conservation of global mean $\zeta$. We now explain this more clearly in the description of SWE in section 4.1.
>
> > Please give the hardware used and simulation time to generate the reference solution and training data. Also, please give the hardware used and GPU-hours for training.
>
> We have added wall clock times and hardware for data generation and training to sections 4.1, 4.2, 5.1 and 5.2.
>
> > Typos: line 68, "to solutions at time via supervised learning" should be "to solutions at time t via supervised learning"? line 256, "Iinitial" should be "Initial", line 347, "perdition" should be "prediction".
>
> We thank the reviewer for the attention to detail and have revised the relevant text.
>
> > Theory. This paper could be improved from the theoretical aspect. For example, the conserved physical quantities are related quantitatively with the solutions, so the losses corresponding to solution prediction and conservation laws are correlated, and maybe a loss bound or convergence rate can be analyzed based on this.
>
> The idea of analyzing errors and error accumulation in terms of constraint violations (physical or symmetry-based) is indeed interesting, though we do not see a path towards obtaining theoretical results on loss bounds or convergence rates. Strictly speaking, there is no loss term for the physical or symmetry constraints during training, as these are imposed as hard constraints on the architecutre. Nonetheless we could forsee quantitatively comparing violations of the constraints in (partially) unconstrained baselines to errors in prediction of future system states. A detailed empirical analysis is beyond our current scope, but we now mention this in our discussion section.

---

### Author Response · Authors · 2024-11-28
**General reply to reviewers**

We wish to thank all 3 reviewers for substantive and constructive comments, which we believe have substantially improved the manuscript. These improvements include several new quantitative results, many errors and fixes, and above all an increase in clarity. We believe this manuscript is now significantly improved in how it conveys the work we carried out, and how this work relates to other work in the field.

---

### Meta-Review · Area_Chair_Fh7N · 2024-12-21

**Metareview:**

This paper studies to incorporate symmetries (the p4m group of translations, reflections, and rotations,) and physical constraints (conservation laws) to improve the neural PDE solvers. Some reviewers acknowledge the paper is well-written and has some value to the community; however, there remain significant concerns, including the limited novelty esp. compared to existing studies, misleading messaging (the authors revised the title based on the suggestion from reviewers), limited scope (only focuses on p4m group), weak experiments, etc. Given these considerations, I believe the paper is not ready for publication in its current state. I suggest the authors significantly revise the paper for future submissions.

**Additional Comments On Reviewer Discussion:**

Reviewer wqWf and Reviewer 8dXp had strong concerns about the paper. The authors provided concise rebuttals but no responses from these two reviewers. I checked the responses and found some major concerns remain. Reviewer qgvH had discussions with the authors, after which some technical details appeared to be clear to the reviewer. However, the main concerns, limited scope and experiments, still stand.

---

### Decision · Program_Chairs · 2025-01-22

Reject